# Screening Feedback for Language Models with Costly Verification

## Abstract

Human feedback is central to modern language-model training, but feedback collection raises an incentive-design problem: platforms must attract useful contributors while limiting harmful or low-quality feedback. We study this problem in a stylized screening model with costly verification. A platform commits to a uniform verification–reward–sanction policy, while users differ in both feedback helpfulness and effective sanction exposure. We characterize when sanction-based screening yields normal separation, in which high-quality contributors participate, and when it yields reverse screening, in which high-quality contributors are deterred. The screening boundary depends on the types' effective expected sanction exposures and is independent of the quadratic cost structure used later for closed-form policy characterizations. Under quadratic verification and enforcement costs, we derive least-cost relaxed policies within the screening regimes, interpreted as weak implementations or as limits of strict $\varepsilon$-implementations when participation constraints bind. We also compare screening against pooling and no participation, and show that platform value need not increase monotonically with the share of high-quality contributors. An illustrative bigram simulation visualizes the theoretical regimes and policy trade-offs; it is not intended as empirical validation of deployed LLM feedback systems. The results caution that uniform penalties in feedback pipelines can backfire when valuable contributors face greater effective exposure to sanctions.

**Keywords:** Costly verification, Incentives, User feedback, Quality control, Platform economics, Language models

## 1 Introduction

Human feedback is a central input to modern language-model training and alignment pipelines. Instruction tuning, preference modeling, evaluation, and reinforcement learning from human feedback all rely on contributors whose judgments or demonstrations help determine which model behaviors are reinforced. Yet feedback collection creates a basic platform-design problem: useful contributors must be attracted and retained, while harmful or low-quality feedback must be limited before it degrades model performance or increases downstream curation costs.

We study this problem in a stylized screening environment with costly verification. A platform commits to a uniform policy $(\rho, R, P)$ applied to all users: a verification rate $\rho$, a reward $R$ for submitting feedback, and a sanction $P$ imposed when verified feedback is harmful. Users differ in two ways. High-type users are more likely than low-type users to submit helpful feedback, and types may also differ in their effective exposure to sanctions. We denote these sanction-exposure parameters by $\phi_H$ and $\phi_L$. They are reduced-form primitives capturing behavioral and institutional channels such as risk aversion, loss aversion, reputational concerns, enforceability, appealability, salience of penalties, and dependence on future platform access.

Our setting is not a direct-revelation mechanism in which users report types and the platform assigns type-contingent contracts. Instead, expertise is difficult to certify ex ante, and the platform uses a uniform product-level policy. Users observe this policy and decide whether to participate. The platform's problem is therefore one of screening and enforcement under costly verification.

The first contribution is to characterize the boundary between normal separation and reverse screening. Normal separation is the intended sorting pattern: high-type users participate and low-type users abstain. Reverse screening is the opposite pattern: low-type users participate while high-type users are deterred. We show that sanction-based screening is governed by the comparison between $\phi_H(1-\eta_H)$ and $\phi_L(1-\eta_L)$, where $\eta_t$ is the probability that type $t$ submits helpful feedback conditional on participation. Normal separation is feasible when $\phi_H(1-\eta_H) < \phi_L(1-\eta_L)$, while reverse screening arises when the inequality is reversed. This boundary follows from participation constraints and does not depend on the quadratic verification and enforcement costs used later for closed-form policy characterizations.

The reverse-screening region identifies a failure mode of sanction-based governance. Penalties intended to deter harmful feedback may instead deter high-quality contributors if those contributors face sufficiently high effective sanction exposure. Thus, stronger enforcement need not improve screening; it can change which users are willing to participate. The result is a caution about uniform penalties, not a recommendation to punish contributors more aggressively.

The second contribution is to characterize verification, reward, and sanction policies within the pure-participation regimes. For tractability and transparency, we derive closed-form rules under quadratic verification and enforcement costs. These policies are least-cost policies, so the participating type's participation constraint may bind. We therefore interpret them as weak implementations under a participation tie-breaking rule, or as limits of strict $\varepsilon$-implementations obtained by adding an arbitrarily small reward slack. Cases with $P^*\rho^* = 0$ are relaxed-boundary solutions rather than exact strict-separation implementations.

Verification is the main policy margin: it improves the value of screened feedback and enables sanctions, but aggregate verification costs are convex in the mass of verified feedback. Rewards are pinned down by participation constraints. Sanctions are useful only when expected sanction collections exceed the induced increase in reward compensation needed to retain the targeted participating type, and they are limited by enforcement and reputational costs. In practical settings, sanctions may also face limited-liability, deposit, appealability, or proportionality constraints; these constraints are discussed as broader-impact and implementation considerations rather than built into the baseline model.

We also compare feasible screening against pooling, in which all users participate, and no participation. This benchmark comparison is important because screening is not automatically globally optimal. Even when normal or reverse screening is feasible, pooling may dominate if deterring one type sacrifices too much useful participation mass or requires costly incentives. For each parameter vector, the platform therefore compares the best feasible one-type screening policy with pooling and no participation.

The third contribution is a conditional non-monotonicity result for population quality. Under endogenous verification and convex verification costs, platform value need not increase monotonically with the share of high-type users. A higher high-type share improves the composition of potential contributors, but it can also increase the mass of users subject to costly verification and incentive payments. Under the quadratic-cost specification, we give conditions under which optimal profit declines over some range of population quality. This is a possibility result rather than a universal prediction.

Finally, we provide an illustrative simulation. A transparent bigram language-model environment generates helpfulness probabilities $\widehat{\eta}_H$ and $\widehat{\eta}_L$, and these same estimated values are used in the macro policy calculations. The exercise is not intended as empirical validation of modern LLM feedback pipelines. Instead, it is a controlled visualization device that shows how the theoretical regimes, pooling comparison, policy instruments, and local non-monotonicity mechanism appear under explicit parameter choices. Reproducibility details, including the random seed, corpus split, preprocessing, update rule, validation protocol, and macro parameter choices, are reported in Appendix D; the full Python script and generated outputs are provided in the supplementary material.

This paper connects work on human feedback and alignment, crowdsourcing and information elicitation, costly verification, platform governance, and behavioral responses to sanctions. Section 2 positions the paper relative to these literatures. The broader implication is that feedback governance cannot be reduced to stronger verification or harsher penalties. Verification, rewards, and sanctions jointly determine who participates, and sanctions may impose costs on good-faith contributors or affect user groups differently.

The remainder of the paper is organized as follows. Section 2 reviews related work. Section 3 presents the model. Section 4 characterizes screening regimes, closed-form policies, the pooling benchmark, and the non-monotonicity result. Section 5 provides illustrative simulations. Section 6 discusses broader-impact considerations. Section 7 concludes.

## 2  Related Work

Our paper connects work on human feedback for language models, crowdsourcing and information elicitation, costly verification, platform governance, and behavioral responses to sanctions. The distinctive feature of our setting is that a platform commits to a uniform policy $(\rho, R, P)$ consisting of verification, rewards, and sanctions. Contributors do not report their types, and the platform does not offer type-contingent menus. Instead, feedback quality is difficult to certify ex ante, costly to verify ex post, and shaped by participation incentives.

**Human feedback and alignment for language models.**  Human feedback is central to instruction tuning, preference modeling, and reinforcement learning from human feedback (Christiano et al., 2017; Stiennon et al., 2020; Ouyang et al., 2022; Bai et al., 2022). Recent work also studies alternatives and refinements to standard RLHF, including direct preference optimization and related preference-based objectives (Rafailov et al., 2023; Azar et al., 2024; Ethayarajh et al., 2024). A complementary literature highlights limitations of feedback-based training, including reward-model overoptimization, reward hacking, and mismatch between measured preferences and desired model behavior (Gao et al., 2023; Skalse et al., 2022; Casper et al., 2023). Our paper does not propose a new alignment algorithm. Instead, it studies the incentive and verification layer that determines which feedback enters such pipelines.

**Crowdsourcing, elicitation, and data quality.**  A large literature studies how to obtain reliable information from heterogeneous contributors. Work on collective intelligence emphasizes that diverse contributors can generate value through complementary information or problem-solving heuristics (Hong & Page, 2004; Aggarwal et al., 2015), while crowdsourcing research studies worker heterogeneity, noisy labels, and incentive design (Dawid & Skene, 1979; Snow et al., 2008; Raykar et al., 2010; Shah & Zhou, 2016). Peer-prediction and information-elicitation mechanisms provide incentives for truthful reports when ground truth is unavailable or costly to observe (Prelec, 2004; Miller et al., 2005; Witkowski & Parkes, 2012). In machine learning, data quality is also addressed through technical tools such as active learning and label-error detection (Settles, 2009; Northcutt et al., 2021). Our setting differs by focusing on participation screening when feedback can be verified ex post but verification is costly and probabilistic.

**Costly verification and auditing.**  The model builds on the economic logic of costly state verification and auditing (Townsend, 1979; Gale & Hellwig, 1985; Kofman & Lawarrée, 1993). In these models, a principal verifies hidden information at a cost and designs incentives around the possibility of audit. We adapt this logic to feedback collection for language models, where the hidden object is feedback helpfulness and verification both improves model value and enables sanctions. The screening boundary in our paper follows from participation constraints and does not require the quadratic cost specification. The closed-form optimal-policy results and the non-monotonicity proposition use quadratic verification and enforcement costs to obtain transparent comparative statics.

**Platform governance and sanctions.**  Our analysis also relates to platform economics and governance. Platforms shape participation and value creation through prices, access rules, reputation systems, moderation, and enforcement (Rochet & Tirole, 2003; Armstrong, 2006; Cabral & Hortacsu, 2010; Tadelis, 2016). Recent work in AI and data markets studies incentive design for data procurement and model training environments (Duetting et al., 2024; Sun et al., 2024). Relative to this literature, we focus on how a uniform reward–verification–sanction policy changes the composition of participating feedback providers. The reverse-screening result shows that enforcement can backfire when high-quality contributors face greater effective sanction exposure than low-quality contributors.

**Effective sanction exposure.** The type-specific parameters $\phi_H$ and $\phi_L$ summarize how formal sanctions translate into effective participation costs. This reduced-form approach is motivated by both behavioral and institutional considerations. Users may differ in risk aversion, loss aversion, reputational concerns, dependence on platform access, or beliefs about enforcement and appealability (Kahneman & Tversky, 1979; Tversky & Kahneman, 1991; Becker, 1968; Polinsky & Shavell, 2000). Thus, $\phi_t$ should not be interpreted only as psychological penalty aversion. It captures the effective burden of sanctions for type $t$, including both perceived costs and institutional features of enforcement.

## 3 Feedback Screening with Heterogeneous Users

We consider a sequential screening game between a platform, interpreted as a language-model provider, and a continuum of potential feedback contributors. Users may submit feedback that can improve or harm the platform's model. The platform cannot observe contributor type ex ante and therefore commits to a uniform verification, reward, and sanction policy. The key friction is that feedback quality can be assessed, but only through costly verification.

The timing is as follows.

1. The platform commits to a uniform policy $(\rho, R, P)$, where $\rho \in [0,1]$ is the verification rate, $R \geq 0$ is the reward for submitting feedback, and $P \geq 0$ is a sanction applied when verified feedback is harmful.

2. Users observe $(\rho, R, P)$ and each user chooses an action $a_i \in \{F, N\}$, where $F$ denotes submitting feedback and $N$ denotes not submitting feedback. Submitting feedback entails a private cost $c > 0$.

3. If $a_i = F$, the user submits a feedback response. The platform verifies the submitted feedback with probability $\rho$. If the feedback is verified and harmful, the platform detects it, excludes it from the verified-feedback update, and applies the sanction.

The game is therefore a sequential commitment game. We focus on subgame-perfect equilibrium outcomes induced by the platform's commitment to $(\rho, R, P)$.

The user's strategic decision in the baseline model is whether to participate, not how much effort to exert or how to strategically choose the content of feedback. Conditional on participation and type, a submitted feedback instance is generated by a type-dependent distribution. Thus, the platform policy screens the composition of participating users, while the conditional helpfulness probabilities defined below are treated as exogenous primitives. This abstraction isolates the participation-screening effect of verification and sanctions. Endogenous effort, strategic content choice, or learning-by-contributors are natural extensions but are outside the baseline model.

### 3.1 Users, feedback, and helpfulness

There is a continuum of users indexed by $i \in [0,1]$. Each user has a private type $t_i \in \{H, L\}$, where $H$ denotes a high type and $L$ denotes a low type. Types are independently distributed with $\mathbb{P}(t_i = H) = \lambda$ and $\mathbb{P}(t_i = L) = 1 - \lambda$. High-type users are more likely to submit helpful feedback than low-type users.

We model the language model through the effect of user feedback on a validation objective. Let $x_i$ denote the prompt or context associated with user $i$'s feedback, and let $y_i$ denote the submitted feedback response or continuation. Thus, feedback in the model is not a numerical score or a binary label; it is a response, continuation, or training instance that the platform may incorporate into model updating.

Formally, conditional on participation and type, the submitted feedback instance $(x_i, y_i)$ is drawn from a type-dependent distribution. The distribution for high types induces a larger probability of validation-improving feedback than the distribution for low types. The model does not require specifying the full feedback-generation distribution; it uses only the induced helpfulness probabilities.

Let $f^0$ denote the baseline model before incorporating user $i$'s feedback, and let $f^{(i)}$ denote the counterfactual model obtained by updating $f^0$ on the feedback instance $(x_i, y_i)$ according to the platform's update rule. We classify a feedback instance as helpful if incorporating it improves expected log-likelihood on a held-out validation distribution $\mathcal{V}$. Formally, define

$$\tilde{\sigma}(x_i, y_i) := \mathbb{E}_{(x', y') \sim \mathcal{V}} \left[ \log f^{(i)}(y' \mid x') - \log f^0(y' \mid x') \right], \qquad r_i := \mathbf{1}\{\tilde{\sigma}(x_i, y_i) > 0\}. \tag{1}$$

Thus, $r_i = 1$ means that the submitted feedback is helpful according to the validation-improvement criterion, while $r_i = 0$ means that it is harmful.

The type-dependent helpfulness probability is

$$\eta_t := \mathbb{P}(r_i = 1 \mid t_i = t, a_i = F) = \mathbb{P}\big(\tilde{\sigma}(x_i, y_i) > 0 \mid t_i = t, a_i = F\big), \qquad t \in \{H, L\}. \tag{2}$$

We assume $\eta_H > \eta_L$, so high-type users are more likely to submit helpful feedback. These probabilities are policy-invariant in the baseline model: changing $(\rho, R, P)$ affects whether each type participates, but not the conditional distribution of feedback quality generated by a participating user of a given type.

The validation-improvement criterion is a reduced-form modeling choice. It is motivated by likelihood-based approaches to language-model adaptation and preference optimization, where improvements in log probabilities or log-probability ratios are central objects. For example, Ranked Choice Preference Optimization (RCPO) (Tang & Feng, 2025) frames preference alignment as maximum likelihood estimation under ranked choice models. In that framework, each prompt $x_i$ is associated with a candidate response set $S_i$ and an observed ranking or top-$k$ choice $\mu_i^k$. A generic ranked choice likelihood takes the form $\sum_i \log g \left( \mu_i^k, S_i, \{r_{\pi_\theta}(x_i, y)\}_{y \in S_i} \right)$, where $g$ is the ranked choice probability model and $r_{\pi_\theta}(x_i, y)$ is a reward or utility score assigned to candidate response $y$ under policy $\pi_\theta$.

This perspective is also connected to the DPO family of preference objectives. In DPO-style formulations, the implicit reward can be written as a log-probability ratio relative to a reference policy:

$$r_{\pi_\theta}(x, y) = \beta \log \frac{\pi_\theta(y \mid x)}{\pi_{\text{ref}}(y \mid x)} + \beta \log Z(x), \tag{3}$$

where $\pi_\theta$ is the current policy, $\pi_{\text{ref}}$ is the reference policy, $\beta > 0$ is a temperature parameter, and $Z(x)$ is a prompt-dependent normalization term. Since $Z(x)$ does not vary across candidate responses for the same prompt, preference learning is driven by differences in log probabilities or log-probability ratios.

We emphasize that the RCPO and DPO discussion is only motivational. None of the screening results below require a particular preference-optimization objective, ranked-choice model, reward parameterization, or update rule. The economic model uses only the induced helpfulness probabilities $\eta_H$ and $\eta_L$. The purpose of the likelihood-based discussion is to explain why validation log-likelihood improvement is a natural reduced-form criterion for classifying whether a submitted feedback response improves model fit.

### 3.2 Verification and sanctions

If user $i$ submits feedback, the platform verifies the instance with probability $\rho$. Let $v_i \sim \text{Bernoulli}(\rho)$ denote the verification indicator. If $v_i = 1$, the platform observes whether the feedback is helpful or harmful. Verified helpful feedback is retained and contributes to model quality. Verified harmful feedback is detected, excluded from the verified-feedback update, and triggers the sanction. Thus, verified harmful feedback does not enter the positive verified-quality term in the platform payoff. If $v_i = 0$, the platform does not observe whether the feedback is helpful or harmful, and the feedback enters through the reduced-form average-quality term introduced below.

Transfers to user $i$ are

$$m_i = \begin{cases} R & \text{if } a_i = F \text{ and } (v_i = 0 \text{ or } r_i = 1), \\ R - P & \text{if } a_i = F \text{ and } v_i = 1 \text{ and } r_i = 0, \\ 0 & \text{if } a_i = N. \end{cases} \tag{4}$$

Equivalently, $m_i = \mathbf{1}\{a_i = F\}\left[R - P \cdot \mathbf{1}\{v_i = 1, r_i = 0\}\right]$.

The sanction $P$ should be interpreted as an enforceable platform sanction rather than an unrestricted monetary fine. In practice, it may correspond to withheld rewards, forfeited deposits, reward clawbacks, reduced access to future feedback tasks, reputation losses, or suspension from an account-based feedback program. Thus, the model is most applicable to settings with repeated interaction or platform control over future payments, deposits, reputation, or access.

The baseline model assumes that the platform can choose $P \geq 0$, with increasing enforcement and reputational costs captured in the platform payoff. In applications, sanctions may be constrained by limited liability, deposit balances, legal rules, or platform-design constraints. Such restrictions can be represented by an upper bound $P \leq \bar{P}$, or by a constraint tying sanctions to available rewards or deposits. Imposing such a cap would truncate the optimal sanction levels derived below, but it would not change the screening boundary between normal separation and reverse screening, which follows from participation constraints and depends on the relative effective expected sanction exposures of the two types.

### 3.3 User participation

User decision-making is modeled through a reduced-form participation condition with type-specific exposure to sanctions. Let $\phi_t > 0$ denote the effective sanction-exposure coefficient for type $t$. This coefficient summarizes how type-$t$ users convert a formal sanction into perceived participation costs. It may reflect perceived severity, salience, risk aversion, loss aversion, reputation concerns, appeal opportunities, enforceability, and outside options.

For a type-$t$ user who submits feedback, the probability of facing a sanction is $q_t \equiv \rho(1 - \eta_t)$. This probability is determined by the verification rate and the type's conditional probability of submitting harmful feedback. Let $\bar{U}$ denote the deterministic base utility from receiving the model response, independent of type. The reduced-form utility of a type-$t$ user is

$$u_t(a_i) = \begin{cases} \bar{U} + R - c - \phi_t P q_t & \text{if } a_i = F, \\ \bar{U} & \text{if } a_i = N. \end{cases} \tag{5}$$

Therefore, a type-$t$ user submits feedback if and only if

$$R - c \geq \phi_t P q_t = \phi_t P \rho(1 - \eta_t). \tag{6}$$

The parameter $\phi_t$ is treated as a reduced-form primitive in the main model. Values above one capture amplified effective exposure to sanctions, for example because of risk aversion, loss aversion, dependence on future platform access, or reputational concerns. Values below one capture attenuated effective exposure, for example because of limited enforceability, imperfect salience, appeal opportunities, weak platform control over future access and payments, or low perceived sanction severity.

Appendix B.1 provides one illustrative mean–variance foundation for this reduced-form utility. That appendix shows how, under a local approximation, risk aversion can amplify the expected sanction term $Pq_t$. This derivation is not a maintained assumption of the main model. The theory below does not require the mean–variance approximation, does not require the approximation error to be zero, and does not impose $\phi_t \geq 1$. Instead, we treat $\phi_t > 0$ as an effective sanction-exposure parameter that can reflect either amplification or attenuation of formal sanctions. Allowing attenuation is important in practice because sanctions may be weakly enforceable, imperfectly salient, appealable, or limited by platform control over future access and payments. It is also important analytically: under the quadratic objective below, imposing $\phi_t \geq 1$ for the participating type would often make sanctions non-profitable and push the relaxed optimal sanction to the boundary $P^* = 0$.

### 3.4 Platform payoff

Let $\theta^{F,H}$ and $\theta^{F,L}$ denote the mass of feedback supplied by high- and low-type users. The corresponding verified and unverified masses are

$$\theta^{v,H} = \rho\,\theta^{F,H}, \qquad \theta^{v,L} = \rho\,\theta^{F,L}, \qquad \theta^u = (1-\rho)(\theta^{F,H} + \theta^{F,L}). \tag{7}$$

For unverified feedback, the platform does not observe whether a specific instance is helpful or harmful. We summarize its effective average quality by reduced-form probabilities $\eta_u^+$ and $\eta_u^-$, where $\eta_u^+, \eta_u^- \in [0,1]$ and $\eta_u^+ + \eta_u^- = 1$. The net term $\eta_u^+ - \eta_u^-$ captures the average effective impact of unverified feedback, possibly after automated filtering, aggregation, downweighting, or noisy incorporation into training.

The unverified-feedback primitive is distinct from the type-level helpfulness probabilities $\eta_H$ and $\eta_L$. The probabilities $\eta_t$ describe the raw probability that a participating type-$t$ user submits validation-improving feedback. By contrast, $\eta_u^+ - \eta_u^-$ summarizes the net value of feedback that is not individually verified and may be processed differently by the platform. It may depend on the participating pool, but in the closed-form policy characterizations below it is treated as a regime-specific primitive. For example, in a special case with no additional filtering or downweighting, the unverified net value for a pool with average helpfulness $\bar{\eta}$ could be represented by $2\bar{\eta} - 1$. The more general formulation allows unverified feedback to be partially filtered, aggregated, or assigned lower training weight.

Model quality improvement is

$$\Delta Q = \gamma_v\big(\eta_H\theta^{v,H} + \eta_L\theta^{v,L}\big) + \gamma_u(\eta_u^+ - \eta_u^-)\theta^u, \tag{8}$$

where $\gamma_v > \gamma_u > 0$ weight verified and unverified feedback. Verified feedback receives the higher weight because verification allows the platform to identify and retain helpful feedback while filtering harmful feedback. Unverified feedback receives the lower weight because its quality is not individually observed and is incorporated only through the reduced-form average-quality term.

The platform chooses $(\rho, R, P)$ to maximize platform payoff, trading off quality improvements, verification costs, reward payments, penalty collections or recoveries, and enforcement or reputational costs. We use a quadratic verification cost, $\frac{k}{2}(\theta^{v,H} + \theta^{v,L})^2$ with $k > 0$, to capture convex review costs and diseconomies of scale. As the verified mass grows, the platform may need to allocate scarcer expert review time, adjudicate more ambiguous cases, or maintain more costly audit infrastructure. The policy-level enforcement and reputational cost of sanctions is $\alpha P^2(\theta^{F,H} + \theta^{F,L})$, where $\alpha > 0$. This term captures the increasing administrative, reputational, contestation, and dispute-resolution costs of maintaining and enforcing more severe sanction schedules for the participating user base.

The platform's objective is

$$\Pi(\rho, R, P) = \underbrace{\gamma_v[\eta_H\theta^{v,H} + \eta_L\theta^{v,L}] + \gamma_u(\eta_u^+ - \eta_u^-)\theta^u}_{\text{Quality improvement}} - \underbrace{\frac{k}{2}(\theta^{v,H} + \theta^{v,L})^2}_{\text{Verification costs}}$$

$$- \underbrace{R(\theta^{F,H} + \theta^{F,L})}_{\text{Reward payments}} + \underbrace{P\rho\big[(1 - \eta_H)\theta^{F,H} + (1 - \eta_L)\theta^{F,L}\big]}_{\text{Penalty collections / recoveries}}$$

$$- \underbrace{\alpha P^2(\theta^{F,H} + \theta^{F,L})}_{\text{Reputational/enforcement costs}}. \tag{9}$$

The payoff terms have the following interpretation. The first line captures model-quality gains from verified helpful feedback and the average net value of unverified feedback. The verification-cost term captures costly review. The reward-payment term is the cost of compensating participating users. The penalty-collection term represents recovered payments, forfeited deposits, clawbacks, or avoided payouts associated with verified harmful feedback. It need not be interpreted as literal fine revenue. The final term captures the administrative, reputational, legal, and dispute-resolution costs of imposing sanctions on the participating user base.

The penalty-collection term should not be interpreted as a normative objective to profit from punishment. It is a reduced-form way to represent monetary recoveries or avoided payments when verified harmful feedback is detected. In settings where sanctions are purely non-monetary or do not generate recoveries for the platform, this term could be reduced or set to zero. The screening-regime boundary derived below would remain governed by participation constraints, although the relaxed optimal sanction levels would change.

The quadratic verification and enforcement cost assumptions are used for tractability and for deriving closed-form policy characterizations with interior verification and sanction trade-offs in Section 4. They should not be interpreted as universal empirical cost laws. The screening-regime boundary itself follows from user participation constraints and does not depend on the quadratic functional form. Under alternative cost specifications, such as linear costs, the same participation-constraint boundary continues to organize normal separation and reverse screening, but the platform's policy problem may collapse to boundary solutions rather than the interior comparative statics studied below.

## 4 Results

We now characterize the platform's screening outcomes and the platform's optimal, or relaxed optimal, policies within the corresponding participation regimes. The analysis proceeds in six steps. First, we clarify the implementation concepts used throughout the section: weak implementation, strict implementation with positive slack, and relaxed conditional optimality. This distinction is important because least-cost policies often place the participating type at its participation boundary. Second, we characterize the boundary between normal separation and reverse screening. This boundary follows from user participation constraints and does not depend on the quadratic cost specification. Third, we explain why mixed participation is a knife-edge outcome rather than a robust screening regime. Fourth, we solve for the platform's relaxed conditional optimum under normal separation using the quadratic verification and enforcement costs in equation 9. Fifth, we present the mirrored relaxed conditional optimum under reverse screening. Finally, we introduce a pooling benchmark in which all users participate, and we state the non-monotonicity result for platform payoff under endogenous verification.

### 4.1 Implementation, Tie-Breaking, and Relaxed Conditional Optima

Before characterizing screening regimes, we distinguish three implementation concepts. This distinction reconciles two features of the model. On the one hand, robust type sorting requires that the excluded type strictly prefers not to participate. On the other hand, the least-cost policy inducing a participating type typically sets that type's participation constraint at equality. The latter equality is a feature of relaxed cost minimization, not a claim that strict implementation requires exact indifference.

Let

$$\Delta_t(\rho, R, P) \equiv (R - c) - \phi_t P \rho (1 - \eta_t), \qquad t \in \{H, L\}, \tag{10}$$

denote the payoff gain from submitting feedback rather than not submitting feedback for a type-$t$ user.

**Weak implementation.** A normal-separation outcome is weakly implemented if high types weakly prefer participation and low types strictly prefer non-participation:

$$\Delta_H(\rho, R, P) \geq 0, \qquad \Delta_L(\rho, R, P) < 0. \tag{11}$$

Similarly, a reverse-screening outcome is weakly implemented if low types weakly prefer participation and high types strictly prefer non-participation:

$$\Delta_L(\rho, R, P) \geq 0, \qquad \Delta_H(\rho, R, P) < 0. \tag{12}$$

When the participating type is exactly indifferent, weak implementation uses the tie-breaking convention that the participating type submits feedback. This convention is used only to describe the least-cost boundary of the implementation set.

**Strict implementation.**    A strict implementation requires the participating type also to strictly prefer participation. Under normal separation, for any policy with $P\rho > 0$ satisfying the normal-separation inequality below, strict participation can be obtained by setting

$$R_\varepsilon = c + \phi_H P\rho(1 - \eta_H) + \varepsilon, \tag{13}$$

where

$$0 < \varepsilon < P\rho\left[\phi_L(1 - \eta_L) - \phi_H(1 - \eta_H)\right]. \tag{14}$$

Then high types strictly participate and low types strictly abstain. The payoff loss relative to the least-cost weak implementation is $\varepsilon\lambda$.

Analogously, under reverse screening, strict implementation can be obtained by setting

$$R_\varepsilon = c + \phi_L P\rho(1 - \eta_L) + \varepsilon, \tag{15}$$

where

$$0 < \varepsilon < P\rho\left[\phi_H(1 - \eta_H) - \phi_L(1 - \eta_L)\right]. \tag{16}$$

Then low types strictly participate and high types strictly abstain. The payoff loss relative to the least-cost weak implementation is $\varepsilon(1 - \lambda)$.

**Relaxed conditional optima.**    The closed-form policy characterizations below solve relaxed conditional problems in which the participating type's constraint is imposed at equality. These policies are useful because they describe the least-cost boundary of the corresponding implementation set. When the relaxed solution has positive screening intensity, $P^*\rho^* > 0$, strict implementation can be obtained by adding an arbitrarily small positive slack $\varepsilon$ to the reward, with arbitrarily small payoff loss.

When the relaxed solution yields $P^* = 0$ or $\rho^* = 0$, the displayed policy does not exactly implement strict type separation, because the reward–penalty screening channel disappears. In such cases, the reported value should be interpreted as a boundary value of the relaxed conditional problem, or where applicable as the supremum approached by strict policies with small positive screening intensity. We therefore distinguish throughout between exact strict implementations and relaxed-boundary policies.

## 4.2   Screening Regimes and the Boundary Condition

The first result characterizes the two pure-participation screening patterns. Normal separation refers to the intended sorting pattern, in which high-type users participate and low-type users abstain. Reverse screening refers to the opposite sorting pattern, in which low-type users participate and high-type users are deterred.

**Theorem 4.1** (Screening regimes)**.** *Suppose sanction-based screening uses positive screening intensity, so $P\rho > 0$. Then:*

1. ***Normal separation.** A normal-separation outcome is weakly implementable if and only if*

$$\phi_H(1 - \eta_H) < \phi_L(1 - \eta_L). \tag{17}$$

   *Equivalently,*

$$\frac{\phi_H}{\phi_L} < \frac{1 - \eta_L}{1 - \eta_H}. \tag{18}$$

   *The least-cost weak implementation sets the high-type participation constraint at equality:*

$$R = c + \phi_H P\rho(1 - \eta_H). \tag{19}$$

   *Moreover, strict implementation is obtained by replacing $R$ with $R_\varepsilon$ as in equation 13 for any $\varepsilon$ satisfying equation 14.*

2. **Reverse screening.** *A reverse-screening outcome is weakly implementable if and only if*

$$\phi_L(1 - \eta_L) < \phi_H(1 - \eta_H). \tag{20}$$

*Equivalently,*

$$\frac{\phi_H}{\phi_L} > \frac{1 - \eta_L}{1 - \eta_H}. \tag{21}$$

*The least-cost weak implementation sets the low-type participation constraint at equality:*

$$R = c + \phi_L P \rho (1 - \eta_L). \tag{22}$$

*Moreover, strict implementation is obtained by replacing $R$ with $R_\varepsilon$ as in equation 15 for any $\varepsilon$ satisfying equation 16.*

*The equality case $\phi_H(1 - \eta_H) = \phi_L(1 - \eta_L)$ is a knife-edge boundary at which strict type sorting through $(\rho, R, P)$ is not possible with positive slack for one type and strict exclusion of the other.*

*Proof.* See Appendices B.2 and B.4. □

*Economic implications.* The boundary condition compares the two types' effective expected sanction exposure per unit of sanction-based screening intensity $P\rho$. Normal separation is feasible when high-type users face a lower effective expected sanction from participation than low-type users. Since low types are more likely to generate harmful feedback, this condition is easier to satisfy when their higher error rate is reinforced by greater effective penalty exposure.

Reverse screening arises when this comparison is reversed. In that case, policies that make low types willing to participate may still deter high types because high types face greater effective sanction exposure. This identifies a failure mode of sanction-based governance: penalties intended to screen out harmful feedback may instead discourage the contributors the platform wants to attract.

The binding reward conditions describe the least-cost boundary of each pure-participation regime. Under normal separation, the reward is set just high enough to compensate high types for their participation cost and expected sanction exposure. Under reverse screening, the reward is set just high enough to induce low-type participation. Strict participation by the included type can be obtained by adding an arbitrarily small positive reward slack, as described in Section 4.1.

Verification and sanctions jointly generate the type-dependent expected costs needed for screening. If either $P = 0$ or $\rho = 0$, this screening channel disappears: both types then face the same reward-only participation margin. Thus, exact strict screening through the reward–penalty channel requires $P\rho > 0$. Closed-form policies below that yield $P^* = 0$ or $\rho^* = 0$ should therefore be read as relaxed-boundary policies rather than exact strict-screening implementations.

### 4.3 Mixed Participation and Non-Robust Indifference

The implementation convention above uses indifference only to describe the least-cost boundary of a pure-participation regime. This is distinct from relying on indifference to sustain a non-degenerate interior participation probability. In a sequential commitment game, interior mixed participation, $p_t \in (0, 1)$, can arise only when type $t$ is exactly indifferent between submitting and not submitting feedback. Such mixing is not robust to small perturbations of the platform policy and is therefore not treated as a separate screening regime.

Then type $t$ chooses $F$ if $\Delta_t > 0$, chooses $N$ if $\Delta_t < 0$, and is indifferent if $\Delta_t = 0$.

**Lemma 4.2** (Interior mixing requires knife-edge indifference)**.** *Fix any policy $(\rho, R, P)$. If $\Delta_t(\rho, R, P) \neq 0$, then type $t$ has a unique pure best response, so any best-response participation probability satisfies $p_t \in \{0, 1\}$. If $\Delta_t(\rho, R, P) = 0$, then type $t$ is indifferent and any $p_t \in [0, 1]$ can be supported by an appropriate tie-breaking or mixing convention.*

*Moreover, such interior mixing is not robust to small policy perturbations. For any $\varepsilon > 0$, setting $R' = R + \varepsilon$ yields $\Delta_t(\rho, R', P) > 0$, while setting $R' = R - \varepsilon$ yields $\Delta_t(\rho, R', P) < 0$. Thus an arbitrarily small reward perturbation selects a pure best response. Consequently, mixed or semi-separating participation can arise only at indifference points and is not a robust screening regime under the platform's commitment policy.*

*Proof.* The payoff gain from participation for type $t$ is

$$\Delta_t(\rho, R, P) = (R - c) - \phi_t P \rho (1 - \eta_t). \tag{23}$$

The user's Stage-2 objective is linear in the participation decision. If $\Delta_t(\rho, R, P) > 0$, participation is the unique best response; if $\Delta_t(\rho, R, P) < 0$, non-participation is the unique best response. Hence strict payoff differences imply $p_t \in \{0, 1\}$.

If $\Delta_t(\rho, R, P) = 0$, the user is indifferent between participation and non-participation, so any mixing probability can be supported by a tie-breaking or mixing convention. However, this indifference is knife-edge. For any $\varepsilon > 0$,

$$\Delta_t(\rho, R + \varepsilon, P) = \varepsilon > 0, \qquad \Delta_t(\rho, R - \varepsilon, P) = -\varepsilon < 0. \tag{24}$$

Thus an arbitrarily small reward perturbation breaks indifference and induces a pure best response. The same logic applies to perturbations of $P$ or $\rho$ whenever these instruments affect the expected penalty term. Therefore, interior mixing requires exact indifference and is not robust to small policy perturbations. $\square$

Thus the mixed-participation condition is a boundary, not a robust screening regime. The equality

$$\phi_H(1 - \eta_H) = \phi_L(1 - \eta_L) \tag{25}$$

implies that the two types have the same effective expected sanction exposure per unit $P\rho$. In particular, any policy satisfying the high-type participation indifference condition

$$R = c + \phi_H P \rho (1 - \eta_H) \tag{26}$$

also makes the low type indifferent, and vice versa. Hence strict type sorting through $(R, P, \rho)$ is impossible on equation 25.

Accordingly, we do not treat mixed or semi-separating outcomes as separate equilibrium regimes. Instead, equation 25 is interpreted as the knife-edge boundary separating the two pure-participation screening regions:

- **Normal separation** when $\phi_H(1 - \eta_H) < \phi_L(1 - \eta_L)$;

- **Reverse screening** when $\phi_H(1 - \eta_H) > \phi_L(1 - \eta_L)$.

Within either strict region, the least-cost weak implementation can be approximated arbitrarily closely by strict implementations whenever $P\rho > 0$, as shown in Section 4.1.

### 4.4 Relaxed Conditional Optimum under Normal Separation

We next characterize the platform's relaxed conditional optimum under normal separation. In this regime, high-type users participate and low-type users abstain, so $\theta^{F,H} = \lambda$, $\theta^{F,L} = 0$, $\theta^{v,H} = \rho\lambda$, $\theta^{v,L} = 0$, and $\theta^u = (1 - \rho)\lambda$. The following result solves the relaxed conditional problem in which the high-type participation constraint is imposed at equality. Strict implementation with positive high-type participation slack is obtained by adding an arbitrarily small $\varepsilon > 0$ to the reward when $P^* \rho^* > 0$.

**Theorem 4.3** (Relaxed conditional optimum under normal separation)**.** *Consider the normal-separation region in Theorem 4.1. Under the quadratic verification and enforcement costs in equation 9, the platform's relaxed conditional optimum is characterized by the following cases.*

**Case 1:** $\phi_H < 1$**. If**

$$k\lambda > \frac{(1-\eta_H)^2(1-\phi_H)^2}{2\alpha}, \tag{27}$$

*then the relaxed conditional optimum is*

$$\rho^* = \min\left\{1, \max\left\{0, \frac{\gamma_v\eta_H - \gamma_u(\eta_u^+ - \eta_u^-)}{k\lambda - \frac{(1-\eta_H)^2(1-\phi_H)^2}{2\alpha}}\right\}\right\}, \tag{28}$$

$$P^* = \frac{\rho^*(1-\eta_H)(1-\phi_H)}{2\alpha}, \tag{29}$$

$$R^* = c + \phi_H P^*\rho^*(1-\eta_H). \tag{30}$$

**Case 2:** $\phi_H \geq 1$**, or** $\phi_H < 1$ **with non-concavity.** *If* $\phi_H \geq 1$*, then penalties are not payoff-improving in the relaxed conditional problem and*

$$\rho^* = \min\left\{1, \max\left\{0, \frac{\gamma_v\eta_H - \gamma_u(\eta_u^+ - \eta_u^-)}{k\lambda}\right\}\right\}, \tag{31}$$

$$P^* = 0, \tag{32}$$

$$R^* = c. \tag{33}$$

*If instead* $\phi_H < 1$ *but condition equation 27 is violated, then the relaxed objective is weakly convex in verification intensity and the optimum occurs at a boundary:*

$$\rho^* \in \{0, 1\}, \tag{34}$$

*with*

$$\rho^* = \begin{cases} 1, & \text{if } \gamma_v\eta_H - \gamma_u(\eta_u^+ - \eta_u^-) - \frac{k\lambda}{2} + \frac{(1-\eta_H)^2(1-\phi_H)^2}{4\alpha} \geq 0, \\ 0, & \text{otherwise}, \end{cases} \tag{35}$$

*and*

$$P^* = \frac{\rho^*(1-\eta_H)(1-\phi_H)}{2\alpha}, \tag{36}$$

$$R^* = c + \phi_H P^*\rho^*(1-\eta_H). \tag{37}$$

*Proof.* See Appendix B.3. □

*Economic implications.* Conditional on normal separation, verification is the platform's main margin of adjustment. In Case 1, penalties are useful because the high type's effective sanction-exposure coefficient is below one. With the high-type participation constraint binding in the relaxed problem, increasing $P$ raises the reward required to keep high types participating by $\phi_H\rho(1-\eta_H)$ per unit of participating high-type mass. However, it also raises expected penalty collections by $\rho(1-\eta_H)$. Hence the net marginal benefit of increasing the penalty before enforcement and reputational costs is $\rho(1-\eta_H)(1-\phi_H)$, which is positive when $\phi_H < 1$. The relaxed optimal penalty therefore rises with verification intensity, while the reward is pinned down by the least-cost participation constraint.

The denominator in equation 28 captures the net curvature of verification. Verification is costly, but it also increases the effectiveness of penalties by raising the probability that harmful feedback is detected and sanctioned. When condition equation 27 holds, this trade-off is concave and yields an interior verification rule, truncated to the feasible interval. When high types are sufficiently penalty-sensitive, penalties are no longer payoff-improving because the induced reward increase weakly exceeds expected penalty collections, so the relaxed conditional problem sets $P^* = 0$ and relies only on verification. If condition equation 27 fails,

the verification objective becomes non-concave, and the relaxed optimum is at a boundary with either no verification or full verification.

Because exact strict separation requires $P\rho > 0$, cases with $P^* = 0$ or $\rho^* = 0$ are not exact strict-separation implementations. They are boundary solutions of the relaxed conditional problem. When $P^*\rho^* > 0$, strict normal separation can be implemented by replacing $R^*$ with $R^* + \varepsilon$ for sufficiently small $\varepsilon > 0$, with payoff loss $\varepsilon\lambda$.

### 4.5 Relaxed Conditional Optimum under Reverse Screening

The reverse-screening policy characterization mirrors the normal-separation case, with the participating type changed from $H$ to $L$ and the participating mass changed from $\lambda$ to $1 - \lambda$. We state the result separately to make the failure mode of sanction-based screening explicit.

**Theorem 4.4** (Relaxed conditional optimum under reverse screening). *Consider the reverse-screening region in Theorem 4.1, where high-type users abstain and low-type users participate. Under the quadratic verification and enforcement costs in equation 9, the platform's relaxed conditional optimum is characterized by the following cases.*

*Case 1: $\phi_L < 1$. If*

$$k(1 - \lambda) > \frac{(1 - \eta_L)^2(1 - \phi_L)^2}{2\alpha}, \tag{38}$$

*then the relaxed conditional optimum is*

$$\rho^* = \min\left\{1, \max\left\{0, \frac{\gamma_v\eta_L - \gamma_u(\eta_u^+ - \eta_u^-)}{k(1 - \lambda) - \frac{(1-\eta_L)^2(1-\phi_L)^2}{2\alpha}}\right\}\right\}, \tag{39}$$

$$P^* = \frac{\rho^*(1 - \eta_L)(1 - \phi_L)}{2\alpha}, \tag{40}$$

$$R^* = c + \phi_L P^* \rho^* (1 - \eta_L). \tag{41}$$

*Case 2: $\phi_L \geq 1$, or $\phi_L < 1$ with non-concavity. If $\phi_L \geq 1$, then penalties are not payoff-improving in the relaxed conditional problem and*

$$\rho^* = \min\left\{1, \max\left\{0, \frac{\gamma_v\eta_L - \gamma_u(\eta_u^+ - \eta_u^-)}{k(1 - \lambda)}\right\}\right\}, \tag{42}$$

$$P^* = 0, \tag{43}$$

$$R^* = c. \tag{44}$$

*If instead $\phi_L < 1$ but condition equation 38 is violated, then the relaxed objective is weakly convex in verification intensity and the optimum occurs at a boundary:*

$$\rho^* \in \{0, 1\}. \tag{45}$$

*The boundary choice is*

$$\rho^* = \begin{cases} 1, & \text{if } \gamma_v\eta_L - \gamma_u(\eta_u^+ - \eta_u^-) - \frac{k(1 - \lambda)}{2} + \frac{(1 - \eta_L)^2(1 - \phi_L)^2}{4\alpha} \geq 0, \\ 0, & \text{otherwise,} \end{cases} \tag{46}$$

*with*

$$P^* = \frac{\rho^*(1 - \eta_L)(1 - \phi_L)}{2\alpha}, \tag{47}$$

$$R^* = c + \phi_L P^* \rho^* (1 - \eta_L). \tag{48}$$

*Proof.* The proof can be found in Appendix B.5. $\square$

*Economic implications.* Conditional on reverse screening, the platform optimizes over a participating pool composed only of low-type users. Verification remains the main adjustment margin: it increases the value of screened feedback and enables penalties, but it is costly when applied to the participating mass $1 - \lambda$.

When $\phi_L < 1$, penalties can be payoff-improving because expected penalty collections exceed the additional reward compensation needed to keep low types participating, before enforcement and reputational costs. The relaxed optimal penalty therefore increases with verification intensity. When $\phi_L \geq 1$, penalties are no longer payoff-improving, so the relaxed conditional problem sets $P^* = 0$ and relies only on verification. If the verification objective is non-concave, the relaxed optimum is at a boundary with either no verification or full verification.

As in the normal-separation case, exact strict reverse screening requires $P\rho > 0$. Boundary solutions with $P^* = 0$ or $\rho^* = 0$ are therefore relaxed-boundary policies rather than exact strict implementations. When $P^*\rho^* > 0$, strict reverse screening can be implemented by replacing $R^*$ with $R^* + \varepsilon$ for sufficiently small $\varepsilon > 0$, with payoff loss $\varepsilon(1 - \lambda)$.

### 4.6 Benchmark: Pooling Participation

A natural benchmark is a pooling policy under which all users participate. This benchmark is useful because the screening results above are conditional on the pure-participation region induced by the primitives $\phi_t$ and $\eta_t$. For a fixed parameter vector, the platform does not freely choose between normal separation and reverse screening: when $\phi_H(1 - \eta_H) < \phi_L(1 - \eta_L)$, the feasible one-type screening pattern is normal separation, while when $\phi_H(1 - \eta_H) > \phi_L(1 - \eta_L)$, it is reverse screening. Pooling, however, can be implemented in either region by setting the reward high enough to satisfy both types' participation constraints.

Thus, for each parameter vector, the platform compares the best feasible relaxed one-type screening policy with the optimal pooling benchmark and no participation. This comparison clarifies that screening is not automatically globally preferred: if deterring one type sacrifices too much useful participation mass, if the quality gap is small, or if verification is expensive, allowing all users to participate may yield a higher payoff.

Under pooling,
$$\theta^{F,H} = \lambda, \qquad \theta^{F,L} = 1 - \lambda, \qquad \theta^u = 1 - \rho.$$

Define the population-average helpfulness probability
$$\bar{\eta}(\lambda) \equiv \lambda\eta_H + (1 - \lambda)\eta_L, \tag{49}$$

and the maximum effective sanction exposure across the two types:
$$M \equiv \max\{\phi_H(1 - \eta_H), \phi_L(1 - \eta_L)\}. \tag{50}$$

The least reward inducing both types to participate is
$$R_{\text{pool}} = c + P\rho M. \tag{51}$$

At this reward, the type with maximal effective sanction exposure is exactly at its participation boundary, while the other type weakly has slack. Strict pooling participation by both types can be obtained by adding an arbitrarily small positive reward slack.

Substituting the pooling masses and the least pooling reward into the platform objective gives the reduced pooling payoff
$$\Pi_{\text{pool}}(\rho, P) = \gamma_u(\eta_u^+ - \eta_u^-) - c + A_{\text{pool}}\rho - \frac{k}{2}\rho^2 + P\rho B_{\text{pool}} - \alpha P^2, \tag{52}$$

where
$$A_{\text{pool}} \equiv \gamma_v\bar{\eta}(\lambda) - \gamma_u(\eta_u^+ - \eta_u^-), \qquad B_{\text{pool}} \equiv 1 - \bar{\eta}(\lambda) - M. \tag{53}$$

The term $A_{\text{pool}}$ is the net marginal value of verification under pooling. The term $B_{\text{pool}}$ is the net marginal benefit of increasing sanctions: expected sanction collections rise by $P\rho[1 - \bar{\eta}(\lambda)]$, while the least reward needed to keep both types participating rises by $P\rho M$. Thus the net sanction term is $P\rho[1 - \bar{\eta}(\lambda) - M]$.

**Proposition 1** (Relaxed optimal pooling benchmark)**.** *The platform's relaxed optimal pooling policy is characterized as follows.*

*If $B_{\text{pool}} \leq 0$, sanctions are not payoff-improving under pooling and*

$$P^*_{\text{pool}} = 0, \qquad \rho^*_{\text{pool}} = \min\left\{1, \max\left\{0, \frac{A_{\text{pool}}}{k}\right\}\right\}, \qquad R^*_{\text{pool}} = c. \tag{54}$$

*If $B_{\text{pool}} > 0$, then for each $\rho$,*

$$P^*_{\text{pool}}(\rho) = \frac{\rho B_{\text{pool}}}{2\alpha}. \tag{55}$$

*When $k > B^2_{\text{pool}}/(2\alpha)$, the relaxed optimal verification rate is*

$$\rho^*_{\text{pool}} = \min\left\{1, \max\left\{0, \frac{A_{\text{pool}}}{k - \frac{B^2_{\text{pool}}}{2\alpha}}\right\}\right\}, \tag{56}$$

*with*

$$P^*_{\text{pool}} = \frac{\rho^*_{\text{pool}} B_{\text{pool}}}{2\alpha}, \qquad R^*_{\text{pool}} = c + P^*_{\text{pool}} \rho^*_{\text{pool}} M. \tag{57}$$

*When $B_{\text{pool}} > 0$ and $k \leq B^2_{\text{pool}}/(2\alpha)$, the relaxed pooling objective is weakly convex in $\rho$, so $\rho^*_{\text{pool}} \in \{0, 1\}$. The boundary choice is*

$$\rho^*_{\text{pool}} = \begin{cases} 1, & \text{if } A_{\text{pool}} - \frac{k}{2} + \frac{B^2_{\text{pool}}}{4\alpha} \geq 0, \\ 0, & \text{otherwise.} \end{cases} \tag{58}$$

*The corresponding $P^*_{\text{pool}}$ and $R^*_{\text{pool}}$ are given by equation 57.*

*Proof.* See Appendix B.6. $\qquad\qquad\square$

Unlike one-type screening, pooling does not require $P\rho > 0$. Therefore pooling policies with $P^*_{\text{pool}} = 0$ or $\rho^*_{\text{pool}} = 0$ can still be exact pooling implementations, provided the reward satisfies both types' participation constraints.

### 4.7 Population Quality and Non-Monotone Payoff

We now state the non-monotonicity result as a theoretical comparative static. The result relies on the quadratic verification and enforcement cost structure and on an interior verification solution in the relevant neighborhood. It should therefore be interpreted as a possibility result: platform payoff need not be monotone in population quality once verification responds endogenously and marginal verification costs rise with the verified mass.

**Proposition 2** (Non-monotonicity under quadratic costs and interior verification)**.** *Maintain the normal-separation region and the quadratic verification and enforcement cost structure in equation 9. Suppose $\phi_H < 1$ and define*

$$N \equiv \gamma_v \eta_H - \gamma_u(\eta_u^+ - \eta_u^-), \qquad D \equiv \frac{(1 - \eta_H)^2(1 - \phi_H)^2}{2\alpha}. \tag{59}$$

*Assume that the interior verification solution is feasible in a neighborhood of $\lambda = 1$, i.e.,*

$$0 < N < k - D. \tag{60}$$

*If the participation cost satisfies*

$$c > \gamma_u(\eta_u^+ - \eta_u^-) - \frac{DN^2}{2(k - D)^2}, \tag{61}$$

*then the relaxed conditional platform payoff $\Pi^*(\lambda)$ under normal separation is not monotonically increasing in $\lambda$. In particular, there exists $\bar{\lambda} \in (0, 1)$ such that*

$$\left. \frac{d\Pi^*}{d\lambda} \right|_{\lambda=\bar{\lambda}} < 0. \tag{62}$$

*Proof.* See Appendix B.7. $\square$

The economic mechanism is a scale effect under normal separation. Since low-type users abstain, an increase in $\lambda$ expands the mass of high-type participants rather than improving the average quality of the participating pool. For small and moderate $\lambda$, this can raise the value of verified feedback. However, because verification costs are convex in the verified mass $\rho\lambda$, a larger participating mass also makes verification increasingly costly at the margin. Under the conditions in Proposition 2, the marginal verification-cost effect can dominate, so the relaxed conditional payoff declines over some range of population quality.

This proposition is not a universal inverted-U result. It is a conditional possibility result within the normal-separation region, under quadratic costs and an interior verification solution. The globally selected policy may also involve pooling, no participation, or boundary policies. Consequently, the simulation section distinguishes the general screening boundary from calibration-specific selected-policy paths.

## 5 Illustrative Simulations

We provide a lightweight simulation to complement the theory. The exercise is not intended as empirical validation of modern LLM feedback pipelines, nor is it intended to establish implementation results beyond the analytical model. Its purpose is narrower: to generate plausible type-specific helpfulness probabilities and to visualize the policy regimes characterized above, including normal separation, reverse screening, pooling, and no participation. This also responds to the concern that the optimal screening policy should be compared against the simplest alternative in which all users participate. Throughout the macro-level exercise below, we explicitly compare feasible screening against pooling and no participation.

The simulation has two layers. First, a micro-level bigram environment produces illustrative estimates of the helpfulness probabilities $\eta_H$ and $\eta_L$. Second, these estimated values are fed directly into the closed-form policies from Section 4. The remaining macro parameters are not estimated from the bigram environment; they are platform-level primitives governing verification value, verification congestion, participation costs, and enforcement costs. We therefore choose them to make the theoretical trade-offs visible, and we report the parameter values used in each figure.

For reproducibility, Appendix D reports the random seed, corpus split and preprocessing choices, bigram update rule, validation criterion, macro parameter settings, generated output files, and the sensitivity grid summarized in Appendix C. The full Python script and generated CSV/figure outputs are provided in the supplementary material.

We use the following implementation convention. The closed-form policies are least-cost policies, so the participating type's participation constraint may bind. Such policies can be read either as weak implementations under a tie-breaking rule favoring participation, or as limits of strict $\varepsilon$-implementations obtained by adding an arbitrarily small positive reward slack. When a displayed solution has $P^*\rho^* = 0$, it is a relaxed-boundary solution and should not be interpreted as an exact strict-separation implementation.

### 5.1 Setup and Calibration

The micro-calibration uses a deliberately simple language-modeling environment. We train a smoothed bigram model on the Brown Corpus (Kučera & Francis, 1967), with a synthetic-corpus fallback when the Brown Corpus is unavailable.[1] A prompt is a previous token, and a feedback item is a proposed next token.

---

[1]The Brown Corpus is used only as a lightweight source of token sequences for the illustrative bigram exercise; the simulation does not rely on any domain-specific property of the corpus.

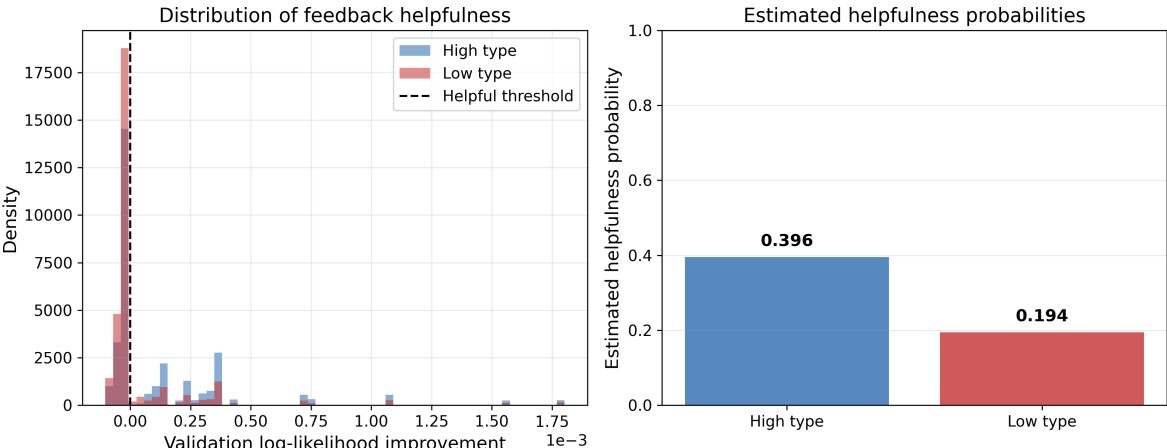

Figure 1: Micro-calibration of type-specific helpfulness. The left panel plots validation log-likelihood improvements from simulated feedback updates. A feedback item is classified as helpful if the improvement is positive. The right panel reports the estimated helpfulness probabilities by type. The estimated values are used directly in the macro-level policy calculations.

To create opportunities for improvement, we first train a baseline bigram model and then degrade a fixed set of prompts by swapping the most likely continuation with a less likely continuation. Special tokens such as [START], [END], and [UNKNOWN] are excluded from the degraded prompt set.

A Monte Carlo iteration proceeds as follows. We sample one prompt from the fixed set of degraded prompts. For that same prompt, we simulate one high-type and one low-type feedback item. The high type submits the ground-truth continuation with probability 0.88, while the low type does so with probability 0.38. These are correctness probabilities inside the toy feedback generator, not the helpfulness probabilities used in the economic model. For each submitted feedback item, we perform a one-row additive update to the flawed bigram transition row and compute the resulting change in validation log-likelihood. A feedback item is classified as helpful if this validation improvement is positive.

Formally, if the current flawed transition row for prompt $x$ is $f(\cdot \mid x)$ and the submitted feedback token is $y$, the updated row is proportional to the old row plus a learning-rate increment on $y$. We then evaluate the validation log-likelihood change using only validation bigrams that begin with $x$. This row-level computation keeps the simulation lightweight while matching the one-step update interpretation.

The theory and simulation treat feedback quality as type-dependent but not strategically chosen in response to the policy. Thus, $\eta_t$ summarizes the probability that a type-$t$ user's feedback is helpful conditional on participation. The platform's policy affects whether a type participates, not the conditional distribution of feedback content. This reduced-form assumption isolates the screening channel. A richer model in which users choose effort or strategically adjust feedback quality in response to $R$, $P$, and $\rho$ is left for future work.

In the macro model, verified harmful feedback is filtered out before it affects model quality. It affects the contributor's payoff through the penalty and affects platform payoff through enforcement costs and any collected penalty revenue, but it does not enter the verified-feedback quality term. Unverified feedback instead enters through the reduced-form net value term $\gamma_u(\eta_u^+ - \eta_u^-)$.

Figure 1 reports the resulting distribution of validation improvements and the estimated helpfulness probabilities. In the run used below, the estimates are $\widehat{\eta}_H = 0.396$ and $\widehat{\eta}_L = 0.194$. Thus, high-type feedback is approximately twice as likely to improve validation performance as low-type feedback. These same estimated values are used in all macro policy simulations below.

Table 1 reports representative examples from the micro-simulation. Rather than selecting only average cases, the table includes high-improvement and high-harm updates, as well as cases in which nominally incorrect feedback improves validation likelihood or nominally correct feedback hurts validation likelihood. These cases

Table 1: Representative feedback updates from the micro-simulation.

| Example type | Prompt | Truth | Flawed model | H correct | H improvement | L correct | L improvement |
|---|---|---|---|---|---|---|---|
| Largest high-type improvement | as | a | he | True | 0.002061 | True | 0.002061 |
| Largest low-type harm | as | a | he | True | 0.002061 | False | -0.001012 |
| Incorrect H feedback helped | designed | to | for | False | 0.000030 | False | 0.000030 |
| Incorrect L feedback helped | beginning | to | of | True | 0.000086 | False | 0.000058 |
| Correct H feedback hurt | trees | , | bright | True | -0.000009 | False | -0.000009 |
| Correct L feedback hurt | heat | of | bright | True | -0.000009 | True | -0.000009 |

illustrate why helpfulness is measured by out-of-sample validation improvement rather than by token-level correctness alone.

## 5.2 Policy Regimes and Optimal Policies

We next feed $\widehat{\eta}_H = 0.396$ and $\widehat{\eta}_L = 0.194$ into the macro policy formulas. These are the only quantities carried over from the micro-calibration. The remaining macro parameters are platform-level primitives: $\gamma_v$ governs the value of verified helpful feedback, $\gamma_u(\eta_u^+ - \eta_u^-)$ summarizes the net value of unverified feedback, $k$ governs verification congestion, $\alpha$ governs enforcement costs, and $c$ is the participation cost. The regime-map calculation uses $\gamma_v = 4.0$, $\gamma_u(\eta_u^+ - \eta_u^-) = 0.4$, $k = 2.0$, $\alpha = 0.8$, and $c = 0.20$. These values are chosen to make the comparison among feasible screening, pooling, and no participation visible.

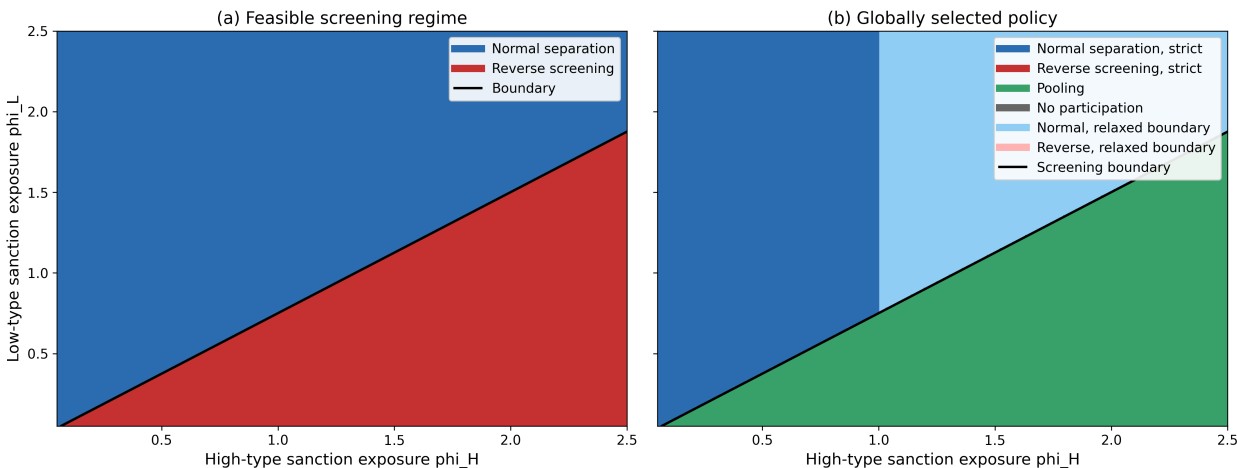

Figure 2: Feasible screening regimes and globally selected policies. Panel (a) shows the screening regime implied by the relative sanction-exposure terms. Panel (b) compares feasible screening against pooling and no participation. The figure illustrates that feasibility of screening does not by itself imply global optimality.

Figure 2 separates two distinct concepts. Panel (a) shows the feasible screening region implied by the sanction-exposure comparison. Normal separation is feasible when the high type has lower effective sanction exposure than the low type; reverse screening is feasible when the inequality is reversed. Panel (b) compares the feasible screening policy, using the relaxed conditional policy formulas where applicable, against pooling and no participation. This distinction is important: a screening regime may be feasible without being globally optimal.

The global policy map also highlights the implementation issue discussed in the theory. If $P^*\rho^* > 0$, the policy has positive screening intensity and can be interpreted as a weak implementation or as the limit of strict $\varepsilon$-implementations. If $P^*\rho^* = 0$, the policy is a relaxed-boundary solution of the closed problem and should not be interpreted as an exact strict-separation implementation. This convention avoids conflating strict separation with limiting cases of the relaxed problem.

Figure 3 fixes $\phi_H = 0.30$ and $\phi_L = 0.80$ and varies the high-type population share $\lambda$. The figure reports the globally selected verification rate, penalty, reward, and profit comparison. To make the non-monotonicity

mechanism visible, this path uses $\gamma_v = 3.4$, $\gamma_u(\eta_u^+ - \eta_u^-) = 0.4$, $k = 5.5$, $\alpha = 0.8$, and $c = 0.35$. The higher congestion and participation-cost parameters make the trade-off between improved population quality and costly verification more pronounced.

The figure also compares the selected policy against feasible screening, pooling, and no participation. At low values of $\lambda$, pooling is selected because high-type users are scarce and screening sacrifices too much participation mass. As $\lambda$ increases, normal separation becomes optimal. Around the regime transition, the selected instruments adjust sharply. The figure should be read as an illustration of the verification-cost channel behind the non-monotonicity result rather than as a universal comparative static.

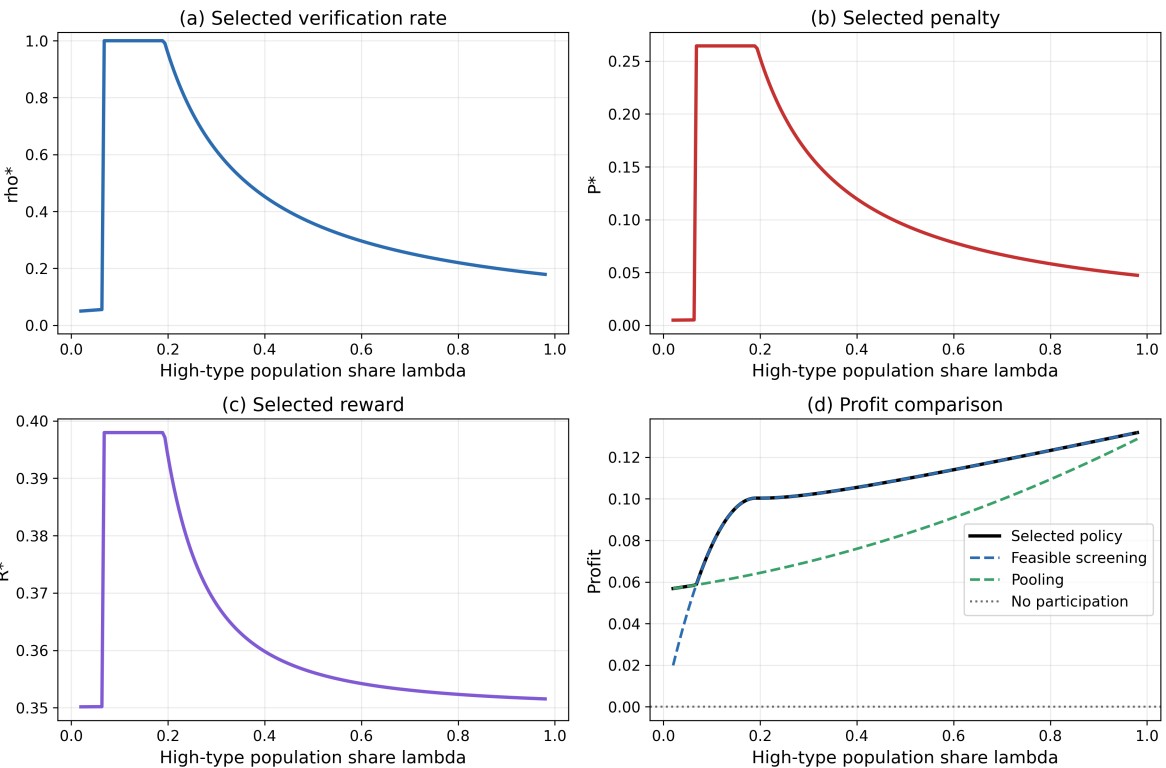

Figure 3: Policy paths and profit comparison under the non-monotonicity calibration. The platform compares feasible screening, pooling, and no participation while varying the high-type population share. The selected verification rate, penalty, and reward adjust sharply around the regime transition. The figure illustrates the verification-cost channel behind the theoretical non-monotonicity result; the pattern is calibration-specific.

Table 2 complements the figures with three representative macro cases. The first is a normal-separation case. The second is near the screening boundary, where policy instruments adjust but positive-screening normal separation remains feasible under the chosen calibration. The third lies in the reverse-screening region. In that case, reverse screening is feasible, but pooling is globally selected. This distinction is useful because the reverse-screening condition characterizes feasible participation incentives, while the platform's global optimum may still be pooling.

### 5.3 Simulation Takeaways

The simulation supports three qualitative takeaways.

First, the micro-calibration produces a clean ranking of helpfulness probabilities, with $\widehat{\eta}_H > \widehat{\eta}_L$. The estimated values are not treated as empirical measurements of LLM feedback pipelines, but they provide plausible inputs for the macro policy exercise.

Table 2: Representative macro policy scenarios. Policies labeled as positive screening have $P^*\rho^* > 0$ under the least-cost formula and should be read as weak implementations or strict $\varepsilon$-implementation limits.

| Scenario | $\lambda$ | $\phi_H$ | $\phi_L$ | Feasible regime | Selected policy | $\rho^*$ | $P^*$ | $R^*$ | Profit |
|---|---|---|---|---|---|---|---|---|---|
| Normal separation | 0.50 | 0.30 | 0.80 | Normal separation | Normal separation (positive screening) | 1.000 | 0.264 | 0.248 | 0.469 |
| Near boundary | 0.50 | 0.70 | 0.552 | Normal separation | Normal separation (positive screening) | 1.000 | 0.113 | 0.248 | 0.446 |
| Reverse screening | 0.50 | 1.50 | 0.20 | Reverse screening | Pooling | 0.390 | 0.000 | 0.200 | 0.352 |

Second, the macro results show why the platform must compare screening against pooling. Even when normal or reverse screening is feasible, it need not be globally optimal. Pooling can dominate when screening sacrifices too much participation mass or when the incentive burden of separation is high. This comparison directly addresses the concern that the proposed optimal policies should be evaluated against the simple alternative of letting all users participate.

Third, the policy-path figure illustrates the non-monotonicity mechanism. Increasing $\lambda$ improves the average quality of potential contributors, but it also changes the mass of users subject to costly verification and incentives. The theorem characterizes parameter regions where this trade-off can generate a local decline in profit; the displayed path is an illustrative calibration of that mechanism.

Appendix C reports a compact grid sensitivity check that varies $\eta_H$, $\eta_L$, $\phi_H$, $\phi_L$, $c$, $k$, and $\alpha$ around the displayed calibration. The sensitivity exercise confirms that the screening boundary is governed by the analytical exposure comparison in Theorem 4.1. By contrast, the inverted-U pattern is conditional rather than universal: it appears for a subset of parameter vectors and can disappear once pooling and no participation are included in the global policy comparison.

Finally, the simulation makes the role of penalty sensitivity visually explicit. The boundary between normal and reverse screening depends on the relative sanction-exposure terms, so changing users' sensitivity to penalties can change which type is easier to retain. The baseline numerical exercise does not impose limited-liability or deposit constraints on $P$. Thus, penalties should be interpreted broadly as platform consequences, such as withheld rewards, reduced access, or reputation costs, rather than necessarily as unrestricted monetary fines. If monetary penalties are used, additional enforceability constraints would be required in practice and could reduce the feasible screening region.

## 6 Broader Impact

This paper studies verification, rewards, and sanctions as platform-governance instruments for human feedback collection. The results should not be interpreted as an endorsement of punitive feedback systems. The main caution is the opposite: sanctions can change who participates, and uniform penalties may deter precisely the contributors whose feedback is most valuable when those contributors face greater effective sanction exposure.

This concern is especially relevant in language-model feedback pipelines, where contributors may differ in expertise, confidence, risk tolerance, reputational concerns, dependence on platform access, and ability to contest platform decisions. A sanction in the model can represent a monetary penalty, withheld reward, account restriction, reputation loss, reduced access to future tasks, or another platform-administered consequence. The sanction-exposure parameters $\phi_H$ and $\phi_L$ should therefore be understood as reduced-form measures of both behavioral penalty sensitivity and institutional exposure. A formally uniform policy can still impose unequal burdens across groups.

The model also highlights risks from noisy or difficult-to-contest verification. False positive findings of harmful feedback may punish good-faith contributors, reduce trust, and exclude users who are more risk-averse or less able to appeal. Before deployment, platforms using spot-check verification and sanctions should evaluate disparate impacts and include safeguards such as transparent verification rules, proportional penalties, limited-liability or deposit constraints, appeal and correction processes, privacy protections for submitted feedback, and non-punitive quality-improvement tools. These safeguards are outside the formal model, but they are natural responses to the reverse-screening mechanism identified here.

A further ethical issue is that the platform objective includes penalty collections. This should be interpreted as a reduced-form accounting term, not as a normative claim that platforms should profit from punishment. In many practical settings, penalties may be better treated as incentive devices subject to proportionality, enforceability, and fairness constraints, rather than as a revenue source. Alternative objectives that maximize model improvement net of operational costs while using penalties only to support incentives are an important direction for future work.

The analysis has several limitations. It is a stylized theoretical model with two user types, reduced-form helpfulness probabilities, and an illustrative simulation rather than an empirical evaluation of a deployed LLM feedback system. The model abstracts from strategic collusion, repeated interaction, multidimensional expertise, demographic heterogeneity, privacy costs, and learning by contributors over time. These simplifications help isolate the screening effect of verification and sanctions, but the results should be read as qualitative guidance for mechanism design rather than direct policy prescriptions.

## 7 Conclusion

This paper studies feedback collection for language models as a screening problem with costly verification. A platform commits to a uniform verification–reward–sanction policy, while contributors differ in feedback helpfulness and effective sanction exposure. The setting captures a practical constraint of many feedback pipelines: expertise and reliability are difficult to certify ex ante, so platforms often rely on product-level rules rather than type-contingent contracts.

The main theoretical result characterizes the boundary between normal separation and reverse screening. Normal separation is feasible when high-type contributors face lower effective expected sanction exposure than low-type contributors; reverse screening arises when this comparison is reversed. This boundary follows from participation constraints and does not depend on the quadratic cost specification used for the closed-form policy formulas. It highlights a governance failure mode: penalties intended to deter harmful feedback may instead deter valuable contributors.

Under quadratic verification and enforcement costs, we characterize least-cost verification, reward, and sanction policies within the pure-participation regimes. Because these policies can place the participating type exactly at its participation constraint, they should be interpreted under the weak-implementation or strict-$\varepsilon$-implementation convention described above. Cases with $P^* \rho^* = 0$ are relaxed-boundary solutions rather than exact strict-separation implementations. This distinction is important for interpreting the closed-form optima and their robustness.

The analysis also compares feasible screening against pooling and no participation. Screening is not automatically globally optimal: pooling can dominate when separation sacrifices too much useful participation or requires costly incentives. In addition, platform value need not increase monotonically with the share of high-quality contributors. With endogenous verification and convex verification costs, a larger high-type population can increase both feedback value and verification burden. The non-monotonicity result is therefore a conditional theoretical mechanism, while the simulation provides an illustrative calibration rather than empirical validation.

The broader lesson is that feedback governance should be evaluated as a joint problem of participation, verification, enforcement, and safeguards. Uniform penalties can backfire when contributors differ in effective sanction exposure, and noisy verification can impose real participation harms on good-faith users. Reliable feedback pipelines therefore require not only better ways to learn from human feedback, but also careful design of the incentives, appeal processes, proportionality constraints, and transparency mechanisms that determine who provides that feedback.

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

## A   Notation Summary

Table 3: Summary of main notation.

| Symbol | Meaning |
| --- | --- |
| *Users, types, and actions* | |
| $i \in [0, 1]$ | User index in the continuum of potential feedback contributors. |
| | *Continued on next page* |

| Symbol | Meaning |
|---|---|
| $t_i \in \{H, L\}$ | User type: high type $H$ or low type $L$. |
| $\lambda$ | Population share of high-type users; the low-type share is $1 - \lambda$. |
| $a_i \in \{F, N\}$ | User action: submit feedback $F$ or not submit $N$. |
| $c$ | Private cost of submitting feedback. |
| $\bar{U}$ | Baseline utility from receiving the model response, independent of feedback participation. |
| *Feedback quality and validation criterion* | |
| $x_i, y_i$ | Prompt/context and submitted feedback response or continuation for user $i$. |
| $f^0$ | Baseline model before incorporating user $i$'s feedback. |
| $f^{(i)}$ | Counterfactual model updated on feedback instance $(x_i, y_i)$. |
| $\mathcal{V}$ | Held-out validation distribution. |
| $\tilde{\sigma}(x_i, y_i)$ | Expected validation log-likelihood improvement from incorporating feedback $(x_i, y_i)$. |
| $r_i \in \{0, 1\}$ | Helpfulness indicator; $r_i = 1$ if feedback improves validation performance. |
| $\eta_t$ | Probability that participating type-$t$ users submit helpful feedback. The model assumes $\eta_H > \eta_L$. |
| *Platform policy, verification, and transfers* | |
| $\rho \in [0, 1]$ | Verification rate chosen by the platform. |
| $R \geq 0$ | Reward paid for submitting feedback. |
| $P \geq 0$ | Sanction applied when verified feedback is harmful. |
| $(\rho^*, R^*, P^*)$ | Optimal or relaxed optimal policy in the relevant regime. |
| $v_i$ | Verification indicator, with $v_i \sim \text{Bernoulli}(\rho)$. |
| $m_i$ | Transfer received by user $i$ under the reward–sanction policy. |
| $\phi_t$ | Effective sanction-exposure coefficient for type $t$. |
| $q_t$ | Probability that type-$t$ feedback is verified and harmful: $q_t = \rho(1 - \eta_t)$. |
| $u_t(a_i)$ | Reduced-form utility of a type-$t$ user from action $a_i$. |
| $\Delta_t(\rho, R, P)$ | Type-$t$ payoff gain from submitting feedback rather than not submitting. |
| *Participation and implementation concepts* | |
| $p_t$ | Participation probability of type $t$, used when discussing mixed participation. |
| $P\rho$ | Sanction-based screening intensity. Strict type separation through sanctions requires $P\rho > 0$. |
| $\phi_t(1 - \eta_t)$ | Type-$t$ effective expected sanction exposure per unit of $P\rho$. |
| $\varepsilon$ | Positive reward slack used to convert least-cost weak implementation into strict implementation. |
| *Feedback masses and platform payoff* | |
| $\theta^{F,t}$ | Mass of type-$t$ users who submit feedback. |
| $\theta^{v,t}$ | Mass of verified type-$t$ feedback, equal to $\rho\theta^{F,t}$. |
| $\theta^u$ | Mass of unverified feedback. |
| $\Delta Q$ | Model-quality improvement from verified and unverified feedback. |
| $\gamma_v$ | Value weight on verified feedback. |
| $\gamma_u$ | Value weight on unverified feedback. |
| $\eta_u^+, \eta_u^-$ | Effective helpfulness and harmfulness components of unverified feedback. |
| $\gamma_u(\eta_u^+ - \eta_u^-)$ | Net value weight on unverified feedback. |
| $k$ | Verification congestion or convex review-cost parameter. |
| $\alpha$ | Enforcement, reputational, or dispute-resolution cost parameter for sanctions. |

| Symbol | Meaning |
|---|---|
| $\Pi(\rho, R, P)$ | Platform payoff under policy $(\rho, R, P)$. |
| *Screening regimes* | |
| Normal separation | Regime in which high types participate and low types abstain. Feasible when $\phi_H(1 - \eta_H) < \phi_L(1 - \eta_L)$. |
| Reverse screening | Regime in which low types participate and high types abstain. Feasible when $\phi_L(1 - \eta_L) < \phi_H(1 - \eta_H)$. |
| Boundary | Knife-edge case in which $\phi_H(1 - \eta_H) = \phi_L(1 - \eta_L)$. |
| *Pooling benchmark* | |
| $\bar{\eta}(\lambda)$ | Population-average helpfulness under pooling: $\bar{\eta}(\lambda) = \lambda\eta_H + (1 - \lambda)\eta_L$. |
| $M$ | Maximum effective sanction exposure under pooling: $M = \max\{\phi_H(1 - \eta_H), \phi_L(1 - \eta_L)\}$. |
| $A_{\text{pool}}$ | Net verification value under pooling: $A_{\text{pool}} = \gamma_v\bar{\eta}(\lambda) - \gamma_u(\eta_u^+ - \eta_u^-)$. |
| $B_{\text{pool}}$ | Net sanction benefit under pooling: $B_{\text{pool}} = 1 - \bar{\eta}(\lambda) - M$. |
| $\Pi_{\text{pool}}$ | Reduced platform payoff under pooling. |
| *Optimal-policy and non-monotonicity terms* | |
| $A_t$ | Type-specific net verification value: $A_t = \gamma_v\eta_t - \gamma_u(\eta_u^+ - \eta_u^-)$. |
| $D_t$ | Type-specific sanction-curvature term: $D_t = (1 - \eta_t)^2(1 - \phi_t)^2/(2\alpha)$. |
| $N$ | Net verification value for high types in the non-monotonicity result: $N = \gamma_v\eta_H - \gamma_u(\eta_u^+ - \eta_u^-)$. |
| $D$ | High-type sanction-curvature term in the non-monotonicity result: $D = (1 - \eta_H)^2(1 - \phi_H)^2/(2\alpha)$. |
| $\Pi^*(\lambda)$ | Platform value or selected policy value as a function of the high-type population share. |
| $\Pi_{[0,1]}$ | Projection operator onto the interval $[0, 1]$, used to express constrained verification rates compactly. |
| *Simulation notation* | |
| $\widehat{\eta}_H, \widehat{\eta}_L$ | Micro-calibrated helpfulness estimates used in the macro simulations. |
| $\omega$ | Bigram feedback update weight in the micro-simulation. |
| $C_{\mathcal{V}}(x, z)$ | Validation bigram count for transition $x \to z$. |

# B  Proofs of Main Results

## B.1  Constant Risk Adjustment Approximation

The main text treats $\phi_t > 0$ as a reduced-form effective sanction-exposure coefficient. This appendix provides one theoretical motivation for the reduced-form utility $\bar{U} + R - c - \phi_t P q_t$. Specifically, a local mean–variance approximation shows how risk aversion can generate a type-specific multiplier on expected sanctions. This derivation is only a motivation for the reduced-form specification. It is not imposed as a restriction in the main model, and the main analysis does not require $\phi_t \geq 1$.

**Lemma B.1** (Constant risk adjustment approximation)**.** *Let $q_t \equiv \rho(1 - \eta_t) \in [0, 1]$. For users who choose $a = F$, mean-variance preferences imply*

$$u_t^{MV}(P, q_t) = \bar{U} + R - c - Pq_t - \psi_t P^2 q_t(1 - q_t),$$

*where $\bar{U}$ is deterministic and $\psi_t \geq 0$ is the type-specific risk aversion parameter.*

*Fix a reference mechanism $(\bar{P}, \bar{q}_t)$ with $\bar{P} \geq 0$ and $\bar{q}_t \in [0, 1]$, and define*

$$\phi_t \equiv 1 + \psi_t \bar{P}(1 - \bar{q}_t).$$

*Then for any $(P, q_t)$,*

$$u_t^{MV}(P, q_t) = \bar{U} + R - c - \phi_t P q_t + \Delta_t(P, q_t),$$

*where the approximation error is given by*

$$\Delta_t(P, q_t) = -\psi_t P q_t \left( P(1 - q_t) - \bar{P}(1 - \bar{q}_t) \right).$$

*Moreover, the error satisfies the bound*

$$|\Delta_t(P, q_t)| \leq \psi_t\, P q_t \left( |P - \bar{P}| + |q_t - \bar{q}_t|\, \max\{P, \bar{P}\} \right).$$

*In particular, when $(P, q_t)$ lies in a neighborhood of $(\bar{P}, \bar{q}_t)$, mean-variance utility is well approximated by the reduced form*

$$u_t^{RF}(P, q_t) \equiv \bar{U} + R - c - \phi_t\, P q_t,$$

*which preserves the endogenous incentive term $P q_t$ while absorbing the local risk premium into a constant type-specific coefficient.*

*Proof.* Starting from mean-variance utility,

$$u_t^{MV}(P, q_t) = \bar{U} + R - c - P q_t - \psi_t P^2 q_t(1 - q_t) = \bar{U} + R - c - P q_t[1 + \psi_t P(1 - q_t)].$$

Add and subtract $\psi_t \bar{P}(1 - \bar{q}_t)$ inside the bracket to obtain

$$u_t^{MV}(P, q_t) = \bar{U} + R - c - P q_t[1 + \psi_t \bar{P}(1 - \bar{q}_t)] - \psi_t P q_t[P(1 - q_t) - \bar{P}(1 - \bar{q}_t)].$$

Defining $\phi_t = 1 + \psi_t \bar{P}(1 - \bar{q}_t)$ yields the stated decomposition, with

$$\Delta_t(P, q_t) = -\psi_t P q_t[P(1 - q_t) - \bar{P}(1 - \bar{q}_t)].$$

The bound follows from the triangle inequality and the facts that $P(1-q_t) \in [0, P]$ and $\bar{P}(1-\bar{q}_t) \in [0, \bar{P}]$. $\square$

Lemma B.1 shows that mean–variance utility can be locally approximated by $\bar{U} + R - c - \phi_t P q_t$. Under this particular motivation, $\phi_t \geq 1$. In the main text, however, $\phi_t > 0$ is treated as a reduced-form effective sanction-exposure coefficient, so the screening analysis does not impose the mean–variance restriction. Values below one are interpreted as attenuated effective exposure to formal sanctions.

## B.2 Proof of Theorem 4.1: Normal-Separation Part

We prove necessity for strict normal separation and sufficiency for weak implementation at the least-cost boundary. We then show how strict implementation is obtained by adding an arbitrarily small positive reward slack.

**Necessity.** Suppose a strict normal-separation outcome exists, with high-type users participating and low-type users abstaining. Thus $p_H = 1$ and $p_L = 0$.

For high types to participate, their participation constraint must hold:

$$R - c - \phi_H P \rho(1 - \eta_H) \geq 0. \tag{PC-H}$$

For low types to abstain strictly, they must strictly prefer non-participation:

$$R - c - \phi_L P \rho(1 - \eta_L) < 0. \tag{NP-L}$$

Equivalently, strict normal separation requires the existence of a reward $R$ such that

$$\phi_H P \rho(1 - \eta_H) \leq R - c < \phi_L P \rho(1 - \eta_L). \tag{63}$$

Strict type sorting through the reward–penalty channel requires $P\rho > 0$. If $P = 0$ or $\rho = 0$, both types face the same reward-only participation margin $R - c$, and strict type separation cannot be implemented through this channel.

Since $P\rho > 0$, dividing equation 63 by $P\rho$ implies

$$\phi_H(1 - \eta_H) < \phi_L(1 - \eta_L). \tag{64}$$

**Sufficiency and least-cost weak implementation.** Suppose

$$\phi_H(1 - \eta_H) < \phi_L(1 - \eta_L). \tag{65}$$

Choose any $P > 0$ and $\rho > 0$, and set the least-cost boundary reward

$$R^0 = c + \phi_H P\rho(1 - \eta_H). \tag{66}$$

Then the high-type payoff gain from participation is

$$\Delta_H(\rho, R^0, P) = R^0 - c - \phi_H P\rho(1 - \eta_H) = 0. \tag{67}$$

Hence high types participate under the tie-breaking convention selecting participation at indifference.

For low types,

$$\begin{aligned}
\Delta_L(\rho, R^0, P) &= R^0 - c - \phi_L P\rho(1 - \eta_L) \\
&= \phi_H P\rho(1 - \eta_H) - \phi_L P\rho(1 - \eta_L) \\
&= P\rho\left[\phi_H(1 - \eta_H) - \phi_L(1 - \eta_L)\right] < 0.
\end{aligned} \tag{68}$$

Thus low types strictly prefer to abstain. Therefore, $R^0$ weakly implements normal separation at the least-cost boundary.

**Strict implementation with positive slack.** To obtain strict high-type participation, replace $R^0$ with

$$R^\varepsilon = c + \phi_H P\rho(1 - \eta_H) + \varepsilon, \tag{69}$$

where

$$0 < \varepsilon < P\rho\left[\phi_L(1 - \eta_L) - \phi_H(1 - \eta_H)\right]. \tag{70}$$

Then

$$\Delta_H(\rho, R^\varepsilon, P) = \varepsilon > 0, \tag{71}$$

so high types strictly prefer to participate. For low types,

$$\Delta_L(\rho, R^\varepsilon, P) = P\rho\left[\phi_H(1 - \eta_H) - \phi_L(1 - \eta_L)\right] + \varepsilon < 0, \tag{72}$$

by equation 70. Hence low types strictly prefer to abstain. Therefore, strict normal separation is implementable whenever equation 65 holds and $P\rho > 0$.

The equivalent ratio form follows by dividing both sides of equation 64 by $\phi_L(1 - \eta_H) > 0$:

$$\frac{\phi_H}{\phi_L} < \frac{1 - \eta_L}{1 - \eta_H}. \tag{73}$$

Finally, fix any $(\rho, P)$ that admits normal separation. If the high-type participation constraint is slack, then the platform can lower $R$ until it binds, preserving weak high-type participation and making low-type participation even less attractive. Since $R$ is a transfer cost for the participating high-type mass, the least-cost weak implementation satisfies

$$R = c + \phi_H P\rho(1 - \eta_H). \tag{74}$$

Strict implementation is obtained from this least-cost boundary by adding any $\varepsilon > 0$ satisfying equation 70. This proves the normal-separation part of Theorem 4.1.

### B.3 Proof of Theorem 4.3

Under normal separation, high-type users participate and low-type users abstain. Hence

$$\theta^{F,H} = \lambda, \qquad \theta^{F,L} = 0, \qquad \theta^{v,H} = \rho\lambda, \qquad \theta^{v,L} = 0, \qquad \theta^u = (1-\rho)\lambda. \qquad (75)$$

The platform's profit function becomes

$$\Pi(\rho, R, P) = \gamma_v \eta_H \rho\lambda + \gamma_u(\eta_u^+ - \eta_u^-)(1-\rho)\lambda - \frac{k}{2}(\rho\lambda)^2 - R\lambda$$
$$+ P\rho(1-\eta_H)\lambda - \alpha P^2\lambda. \qquad (76)$$

By the normal-separation proof, the least-cost weak implementation sets the high-type participation constraint at equality:

$$R = c + \phi_H P\rho(1-\eta_H). \qquad (77)$$

Strict normal separation is obtained by replacing this reward with $R + \varepsilon$, where $\varepsilon > 0$ satisfies equation 70. The resulting payoff loss relative to the relaxed least-cost boundary is $\varepsilon\lambda$, which can be made arbitrarily small. The closed-form policy below therefore characterizes the relaxed conditional optimum, or equivalently the supremum approached by strict $\varepsilon$-implementations when $P\rho > 0$.

Substituting equation 77 into equation 76 yields the reduced objective

$$\Pi(\rho, P) = \lambda \left[ \gamma_v \eta_H \rho + \gamma_u(\eta_u^+ - \eta_u^-)(1-\rho) - \frac{k\lambda}{2}\rho^2 - c \right]$$
$$+ \lambda P\rho(1-\eta_H)(1-\phi_H) - \lambda\alpha P^2. \qquad (78)$$

The choice variables satisfy $\rho \in [0, 1]$ and $P \geq 0$.

**Case 2(i): $\phi_H \geq 1$.** When $\phi_H \geq 1$, we have $1 - \phi_H \leq 0$. Hence, for any fixed $\rho \geq 0$, the penalty-dependent component

$$\lambda P\rho(1-\eta_H)(1-\phi_H) - \lambda\alpha P^2 \qquad (79)$$

is weakly decreasing in $P$ over $P \geq 0$. Therefore, the relaxed optimal penalty is

$$P^* = 0. \qquad (80)$$

With $P^* = 0$, maximizing equation 78 over $\rho \in [0, 1]$ gives

$$\left. \frac{\partial\Pi}{\partial\rho} \right|_{P=0} = \lambda \left[ \gamma_v \eta_H - \gamma_u(\eta_u^+ - \eta_u^-) - k\lambda\rho \right]. \qquad (81)$$

Thus the relaxed optimal verification rate is

$$\rho^* = \min\left\{ 1, \max\left\{ 0, \frac{\gamma_v \eta_H - \gamma_u(\eta_u^+ - \eta_u^-)}{k\lambda} \right\} \right\}. \qquad (82)$$

The reward is

$$R^* = c, \qquad (83)$$

which follows from equation 77 and $P^* = 0$. This establishes Case 2(i). Because $P^* = 0$, this case is a relaxed-boundary solution rather than an exact strict-screening implementation.

**Case 1 and Case 2(ii): $\phi_H < 1$.** When $\phi_H < 1$, we have $1 - \phi_H > 0$. For any fixed $\rho \in [0, 1]$, the reduced objective equation 78 is a strictly concave quadratic in $P$. The first-order condition is

$$\frac{\partial \Pi}{\partial P} = \lambda \left[ \rho(1 - \eta_H)(1 - \phi_H) - 2\alpha P \right] = 0. \tag{84}$$

Therefore,

$$P^*(\rho) = \frac{\rho(1 - \eta_H)(1 - \phi_H)}{2\alpha}. \tag{85}$$

Substituting equation 85 into equation 78 gives

$$\Pi(\rho, P^*(\rho)) = \lambda \left[ \gamma_u(\eta_u^+ - \eta_u^-) - c + \rho \left( \gamma_v \eta_H - \gamma_u(\eta_u^+ - \eta_u^-) \right) - \frac{k\lambda}{2}\rho^2 \right]$$
$$+ \lambda \frac{\rho^2(1 - \eta_H)^2(1 - \phi_H)^2}{4\alpha}. \tag{86}$$

Equivalently, up to a $\rho$-independent constant,

$$\Pi(\rho, P^*(\rho)) = \text{const} + \lambda \left( \gamma_v \eta_H - \gamma_u(\eta_u^+ - \eta_u^-) \right) \rho - \frac{\lambda}{2} \left( k\lambda - \frac{(1 - \eta_H)^2(1 - \phi_H)^2}{2\alpha} \right) \rho^2. \tag{87}$$

**Case 1: concave region.** Suppose

$$k\lambda > \frac{(1 - \eta_H)^2(1 - \phi_H)^2}{2\alpha}. \tag{88}$$

Then equation 87 is strictly concave in $\rho$. The unconstrained maximizer solves

$$\lambda \left( \gamma_v \eta_H - \gamma_u(\eta_u^+ - \eta_u^-) \right) - \lambda \left( k\lambda - \frac{(1 - \eta_H)^2(1 - \phi_H)^2}{2\alpha} \right) \rho = 0. \tag{89}$$

Thus

$$\tilde{\rho} = \frac{\gamma_v \eta_H - \gamma_u(\eta_u^+ - \eta_u^-)}{k\lambda - \frac{(1 - \eta_H)^2(1 - \phi_H)^2}{2\alpha}}. \tag{90}$$

Imposing $\rho \in [0, 1]$ gives

$$\rho^* = \min \left\{ 1, \max \left\{ 0, \frac{\gamma_v \eta_H - \gamma_u(\eta_u^+ - \eta_u^-)}{k\lambda - \frac{(1 - \eta_H)^2(1 - \phi_H)^2}{2\alpha}} \right\} \right\}. \tag{91}$$

The corresponding penalty and reward are

$$P^* = \frac{\rho^*(1 - \eta_H)(1 - \phi_H)}{2\alpha}, \tag{92}$$
$$R^* = c + \phi_H P^* \rho^* (1 - \eta_H). \tag{93}$$

This proves Case 1. If $P^*\rho^* > 0$, strict normal separation is obtained by adding an arbitrarily small reward slack satisfying equation 70; if $P^*\rho^* = 0$, the policy is a relaxed-boundary solution.

**Case 2(ii): non-concave region.** Now suppose $\phi_H < 1$ and

$$k\lambda - \frac{(1 - \eta_H)^2(1 - \phi_H)^2}{2\alpha} \leq 0. \tag{94}$$

Then equation 87 is weakly convex in $\rho$. Therefore, a maximizer over the compact interval $[0, 1]$ occurs at a boundary:

$$\rho^* \in \{0, 1\}. \tag{95}$$

Using equation 86, the boundary-value difference is

$$\Pi(1, P^*(1)) - \Pi(0, P^*(0)) = \lambda \left[ \gamma_v \eta_H - \gamma_u(\eta_u^+ - \eta_u^-) - \frac{k\lambda}{2} + \frac{(1 - \eta_H)^2 (1 - \phi_H)^2}{4\alpha} \right]. \tag{96}$$

Hence

$$\rho^* = \begin{cases} 1, & \text{if } \gamma_v \eta_H - \gamma_u(\eta_u^+ - \eta_u^-) - \dfrac{k\lambda}{2} + \dfrac{(1 - \eta_H)^2 (1 - \phi_H)^2}{4\alpha} \geq 0, \\ 0, & \text{otherwise.} \end{cases} \tag{97}$$

Given $\rho^*$, the penalty and reward are

$$P^* = \frac{\rho^*(1 - \eta_H)(1 - \phi_H)}{2\alpha}, \tag{98}$$

$$R^* = c + \phi_H P^* \rho^*(1 - \eta_H). \tag{99}$$

This proves Case 2(ii) and completes the proof.

When the relaxed solution yields $P^* = 0$ or $\rho^* = 0$, it should be interpreted as a boundary solution of the relaxed conditional problem rather than as an exact strict-screening implementation, because exact strict normal separation requires $P\rho > 0$.

### B.4 Proof of Theorem 4.1: Reverse-Screening Part

We prove both necessity and sufficiency of the reverse-screening condition.

Fix any policy $(\rho, R, P)$. A type $t \in \{H, L\}$ participates if and only if

$$R - c \geq \phi_t P \rho (1 - \eta_t). \tag{100}$$

A strict reverse-screening outcome, with low-type users participating and high-type users abstaining, requires

$$R - c \geq \phi_L P \rho (1 - \eta_L), \tag{PC-L}$$
$$R - c < \phi_H P \rho (1 - \eta_H). \tag{NP-H}$$

Equivalently, there must exist a reward $R$ such that

$$\phi_L P \rho (1 - \eta_L) \leq R - c < \phi_H P \rho (1 - \eta_H). \tag{101}$$

Strict type sorting through the sanction channel requires $P\rho > 0$.

**Necessity.** If strict reverse screening is implementable, then equation 101 holds for some $(\rho, R, P)$ with $P\rho > 0$. Dividing by $P\rho$ gives

$$\phi_L(1 - \eta_L) < \phi_H(1 - \eta_H). \tag{102}$$

**Sufficiency.** Suppose

$$\phi_L(1 - \eta_L) < \phi_H(1 - \eta_H). \tag{103}$$

Choose any $P > 0$ and $\rho > 0$. Then the interval

$$[c + \phi_L P \rho (1 - \eta_L), \ c + \phi_H P \rho (1 - \eta_H)) \tag{104}$$

is nonempty. Pick any $R$ in this interval. Then low types participate and high types strictly abstain, so strict reverse screening is implementable.

The equivalent ratio form follows by dividing both sides of equation 102 by $\phi_L(1 - \eta_H) > 0$:

$$\frac{\phi_H}{\phi_L} > \frac{1 - \eta_L}{1 - \eta_H}. \tag{105}$$

Finally, fix any $(\rho, P)$ that admits strict reverse screening. If the low-type participation constraint is slack, then the platform can lower $R$ until it binds, preserving low-type participation and making high-type participation weakly less attractive. Since $R$ is a transfer cost for the participating low-type mass, any platform-optimal policy implementing reverse screening satisfies

$$R = c + \phi_L P \rho(1 - \eta_L). \tag{106}$$

This proves the reverse-screening part of Theorem 4.1.

## B.5   Proof of Theorem 4.4

Under reverse screening, low-type users participate and high-type users abstain. Hence

$$\theta^{F,L} = 1 - \lambda, \qquad \theta^{F,H} = 0, \qquad \theta^{v,L} = \rho(1 - \lambda), \qquad \theta^{v,H} = 0, \qquad \theta^u = (1 - \rho)(1 - \lambda). \tag{107}$$

The platform's profit function becomes

$$\Pi(\rho, R, P) = \gamma_v \eta_L \rho(1 - \lambda) + \gamma_u(\eta_u^+ - \eta_u^-)(1 - \rho)(1 - \lambda) - \frac{k}{2}\left(\rho(1 - \lambda)\right)^2$$
$$- R(1 - \lambda) + P\rho(1 - \eta_L)(1 - \lambda) - \alpha P^2(1 - \lambda). \tag{108}$$

By the reverse-screening proof, any platform-optimal policy implementing reverse screening satisfies the binding low-type participation constraint:

$$R = c + \phi_L P \rho(1 - \eta_L). \tag{109}$$

Substituting equation 109 into equation 108 yields the reduced objective

$$\Pi(\rho, P) = (1 - \lambda)\left[\gamma_v \eta_L \rho + \gamma_u(\eta_u^+ - \eta_u^-)(1 - \rho) - \frac{k(1 - \lambda)}{2}\rho^2 - c\right]$$
$$+ (1 - \lambda)P\rho(1 - \eta_L)(1 - \phi_L) - (1 - \lambda)\alpha P^2, \tag{110}$$

with $\rho \in [0, 1]$ and $P \geq 0$.

**Case 2(i): $\phi_L \geq 1$.**   When $\phi_L \geq 1$, we have $1 - \phi_L \leq 0$. Hence, for any fixed $\rho \geq 0$, the penalty-dependent component

$$(1 - \lambda)\left[P\rho(1 - \eta_L)(1 - \phi_L) - \alpha P^2\right] \tag{111}$$

is weakly decreasing in $P$ over $P \geq 0$. Therefore, the relaxed optimal penalty is

$$P^* = 0. \tag{112}$$

With $P^* = 0$, maximizing equation 110 over $\rho \in [0, 1]$ gives

$$\left.\frac{\partial \Pi}{\partial \rho}\right|_{P=0} = (1 - \lambda)\left[\gamma_v \eta_L - \gamma_u(\eta_u^+ - \eta_u^-) - k(1 - \lambda)\rho\right]. \tag{113}$$

Thus the relaxed optimal verification rate is

$$\rho^* = \min\left\{1, \max\left\{0, \frac{\gamma_v \eta_L - \gamma_u(\eta_u^+ - \eta_u^-)}{k(1 - \lambda)}\right\}\right\}. \tag{114}$$

The reward is

$$R^* = c, \tag{115}$$

which follows from equation 109 and $P^* = 0$. This establishes Case 2(i).

**Case 1 and Case 2(ii): $\phi_L < 1$.** When $\phi_L < 1$, we have $1 - \phi_L > 0$. For any fixed $\rho \in [0,1]$, equation 110 is a strictly concave quadratic in $P$. The first-order condition is

$$\frac{\partial \Pi}{\partial P} = (1 - \lambda)\left[\rho(1 - \eta_L)(1 - \phi_L) - 2\alpha P\right] = 0. \tag{116}$$

Therefore,

$$P^*(\rho) = \frac{\rho(1 - \eta_L)(1 - \phi_L)}{2\alpha}. \tag{117}$$

Substituting equation 117 into equation 110 gives

$$\Pi(\rho, P^*(\rho)) = (1 - \lambda)\left[\gamma_u(\eta_u^+ - \eta_u^-) - c + \rho\left(\gamma_v \eta_L - \gamma_u(\eta_u^+ - \eta_u^-)\right) - \frac{k(1 - \lambda)}{2}\rho^2\right]$$
$$+ (1 - \lambda)\frac{\rho^2(1 - \eta_L)^2(1 - \phi_L)^2}{4\alpha}. \tag{118}$$

Equivalently, up to a $\rho$-independent constant,

$$\Pi(\rho, P^*(\rho)) = \text{const} + (1 - \lambda)\left(\gamma_v \eta_L - \gamma_u(\eta_u^+ - \eta_u^-)\right)\rho - \frac{1 - \lambda}{2}\left(k(1 - \lambda) - \frac{(1 - \eta_L)^2(1 - \phi_L)^2}{2\alpha}\right)\rho^2. \tag{119}$$

**Case 1: concave region.** Suppose

$$k(1 - \lambda) > \frac{(1 - \eta_L)^2(1 - \phi_L)^2}{2\alpha}. \tag{120}$$

Then equation 119 is strictly concave in $\rho$. The unconstrained maximizer is

$$\tilde{\rho} = \frac{\gamma_v \eta_L - \gamma_u(\eta_u^+ - \eta_u^-)}{k(1 - \lambda) - \frac{(1 - \eta_L)^2(1 - \phi_L)^2}{2\alpha}}. \tag{121}$$

Imposing $\rho \in [0,1]$ gives

$$\rho^* = \min\left\{1, \max\left\{0, \frac{\gamma_v \eta_L - \gamma_u(\eta_u^+ - \eta_u^-)}{k(1 - \lambda) - \frac{(1 - \eta_L)^2(1 - \phi_L)^2}{2\alpha}}\right\}\right\}. \tag{122}$$

The corresponding penalty and reward are

$$P^* = \frac{\rho^*(1 - \eta_L)(1 - \phi_L)}{2\alpha}, \tag{123}$$

$$R^* = c + \phi_L P^* \rho^* (1 - \eta_L). \tag{124}$$

This proves Case 1.

**Case 2(ii): non-concave region.** Now suppose $\phi_L < 1$ and

$$k(1 - \lambda) - \frac{(1 - \eta_L)^2(1 - \phi_L)^2}{2\alpha} \leq 0. \tag{125}$$

Then equation 119 is weakly convex in $\rho$. Therefore, a maximizer over the compact interval $[0,1]$ occurs at a boundary:

$$\rho^* \in \{0, 1\}. \tag{126}$$

Using equation 118, the boundary-value difference is

$$\Pi(1, P^*(1)) - \Pi(0, P^*(0)) = (1 - \lambda)\left[\gamma_v \eta_L - \gamma_u(\eta_u^+ - \eta_u^-) - \frac{k(1-\lambda)}{2} + \frac{(1-\eta_L)^2(1-\phi_L)^2}{4\alpha}\right]. \tag{127}$$

Hence

$$\rho^* = \begin{cases} 1, & \text{if } \gamma_v \eta_L - \gamma_u(\eta_u^+ - \eta_u^-) - \dfrac{k(1-\lambda)}{2} + \dfrac{(1-\eta_L)^2(1-\phi_L)^2}{4\alpha} \geq 0, \\ 0, & \text{otherwise.} \end{cases} \tag{128}$$

Given $\rho^*$, the penalty and reward are

$$P^* = \frac{\rho^*(1-\eta_L)(1-\phi_L)}{2\alpha}, \tag{129}$$

$$R^* = c + \phi_L P^* \rho^* (1 - \eta_L). \tag{130}$$

This proves Case 2(ii) and completes the proof.

When the relaxed solution yields $P^* = 0$ or $\rho^* = 0$, it should be interpreted as a boundary solution of the relaxed conditional problem rather than as an exact strict-screening implementation, because exact strict reverse screening requires $P\rho > 0$.

### B.6 Proof of Proposition 1

Under pooling, both types participate. Hence

$$\theta^{F,H} = \lambda, \qquad \theta^{F,L} = 1 - \lambda, \qquad \theta^{v,H} = \rho\lambda, \qquad \theta^{v,L} = \rho(1-\lambda), \qquad \theta^u = 1 - \rho. \tag{131}$$

The least reward inducing both types to participate is

$$R_{\text{pool}} = c + P\rho M, \tag{132}$$

$$M = \max\left\{\phi_H(1 - \eta_H), \phi_L(1 - \eta_L)\right\}. \tag{133}$$

Substituting these quantities into the platform objective and using the definitions of $\bar{\eta}(\lambda)$, $A_{\text{pool}}$, and $B_{\text{pool}}$ from Section 4.6 gives

$$\Pi_{\text{pool}}(\rho, P) = \gamma_u(\eta_u^+ - \eta_u^-) - c + A_{\text{pool}}\rho - \frac{k}{2}\rho^2 + P\rho B_{\text{pool}} - \alpha P^2, \tag{134}$$

with $\rho \in [0, 1]$ and $P \geq 0$.

For fixed $\rho$, the derivative with respect to $P$ is

$$\frac{\partial \Pi_{\text{pool}}}{\partial P} = \rho B_{\text{pool}} - 2\alpha P. \tag{135}$$

**Case 1: $B_{\text{pool}} \leq 0$.** If $B_{\text{pool}} \leq 0$, then for any fixed $\rho \geq 0$, the objective is weakly decreasing in $P$ over $P \geq 0$. Therefore,

$$P_{\text{pool}}^* = 0. \tag{136}$$

The remaining one-dimensional problem is

$$\max_{\rho \in [0,1]}\left\{\gamma_u(\eta_u^+ - \eta_u^-) - c + A_{\text{pool}}\rho - \frac{k}{2}\rho^2\right\}. \tag{137}$$

The unconstrained maximizer is $A_{\text{pool}}/k$. Imposing $\rho \in [0, 1]$ gives

$$\rho_{\text{pool}}^* = \min\left\{1, \max\left\{0, \frac{A_{\text{pool}}}{k}\right\}\right\}. \tag{138}$$

Finally,

$$R_{\text{pool}}^* = c \tag{139}$$

follows from equation 132 and $P_{\text{pool}}^* = 0$. This proves the case $B_{\text{pool}} \leq 0$.

**Case 2:** $B_{\text{pool}} > 0$. Now suppose $B_{\text{pool}} > 0$. For each fixed $\rho \in [0,1]$, the objective is strictly concave in $P$, and the first-order condition equation 135 gives

$$P^*_{\text{pool}}(\rho) = \frac{\rho B_{\text{pool}}}{2\alpha}. \tag{140}$$

Substituting equation 140 into equation 134 gives, up to a $\rho$-independent constant,

$$\Pi_{\text{pool}}(\rho, P^*_{\text{pool}}(\rho)) = \text{const} + A_{\text{pool}}\rho - \frac{1}{2}\left(k - \frac{B^2_{\text{pool}}}{2\alpha}\right)\rho^2. \tag{141}$$

If

$$k > \frac{B^2_{\text{pool}}}{2\alpha}, \tag{142}$$

then equation 141 is strictly concave in $\rho$. The unconstrained maximizer is

$$\tilde{\rho}_{\text{pool}} = \frac{A_{\text{pool}}}{k - \frac{B^2_{\text{pool}}}{2\alpha}}. \tag{143}$$

Imposing $\rho \in [0,1]$ gives

$$\rho^*_{\text{pool}} = \min\left\{1, \max\left\{0, \frac{A_{\text{pool}}}{k - \frac{B^2_{\text{pool}}}{2\alpha}}\right\}\right\}. \tag{144}$$

The corresponding penalty and reward are

$$P^*_{\text{pool}} = \frac{\rho^*_{\text{pool}} B_{\text{pool}}}{2\alpha}, \tag{145}$$

$$R^*_{\text{pool}} = c + P^*_{\text{pool}}\rho^*_{\text{pool}}M. \tag{146}$$

If

$$k \leq \frac{B^2_{\text{pool}}}{2\alpha}, \tag{147}$$

then equation 141 is weakly convex in $\rho$. Therefore, a maximizer over $[0,1]$ occurs at a boundary:

$$\rho^*_{\text{pool}} \in \{0,1\}. \tag{148}$$

Comparing the two boundary values gives

$$\Pi_{\text{pool}}(1, P^*_{\text{pool}}(1)) - \Pi_{\text{pool}}(0, P^*_{\text{pool}}(0)) = A_{\text{pool}} - \frac{k}{2} + \frac{B^2_{\text{pool}}}{4\alpha}. \tag{149}$$

Hence

$$\rho^*_{\text{pool}} = \begin{cases} 1, & \text{if } A_{\text{pool}} - \frac{k}{2} + \frac{B^2_{\text{pool}}}{4\alpha} \geq 0, \\ 0, & \text{otherwise.} \end{cases} \tag{150}$$

The corresponding penalty and reward are again

$$P^*_{\text{pool}} = \frac{\rho^*_{\text{pool}} B_{\text{pool}}}{2\alpha}, \tag{151}$$

$$R^*_{\text{pool}} = c + P^*_{\text{pool}}\rho^*_{\text{pool}}M. \tag{152}$$

This completes the proof of Proposition 1.

### B.7 Proof of Proposition 2

Work under normal separation, so $\theta^{F,H} = \lambda$ and $\theta^{F,L} = 0$. Consider the interior case with $\phi_H < 1$. After substituting the binding high-type participation constraint, the reduced objective is

$$\Pi(\rho, P; \lambda) = \lambda \left[ \gamma_v \eta_H \rho + \gamma_u (\eta_u^+ - \eta_u^-)(1 - \rho) - c \right] - \frac{k}{2} (\rho \lambda)^2$$
$$+ \lambda P \rho (1 - \eta_H)(1 - \phi_H) - \alpha P^2 \lambda. \tag{153}$$

For fixed $\rho$, the optimal penalty is

$$P^*(\rho) = \frac{\rho(1 - \eta_H)(1 - \phi_H)}{2\alpha}. \tag{154}$$

Define

$$N \equiv \gamma_v \eta_H - \gamma_u (\eta_u^+ - \eta_u^-), \tag{155}$$
$$D \equiv \frac{(1 - \eta_H)^2 (1 - \phi_H)^2}{2\alpha}. \tag{156}$$

Substituting equation 154 into equation 153 gives

$$\Pi(\rho, P^*(\rho); \lambda) = \lambda \left[ \gamma_u (\eta_u^+ - \eta_u^-) - c \right] + \lambda N \rho - \frac{\lambda}{2} (k\lambda - D) \rho^2. \tag{157}$$

When the interior verification solution is feasible, the first-order condition gives

$$\rho^*(\lambda) = \frac{N}{k\lambda - D}. \tag{158}$$

The assumption $0 < N < k - D$ implies $\rho^*(1) \in (0, 1)$. Hence, by continuity, the interior expression is valid for all $\lambda$ in some neighborhood of 1.

Substituting equation 158 into equation 157 gives

$$\Pi^*(\lambda) = \lambda \left[ \gamma_u (\eta_u^+ - \eta_u^-) - c \right] + \frac{\lambda N^2}{2(k\lambda - D)}. \tag{159}$$

Differentiating with respect to $\lambda$ yields

$$\frac{d\Pi^*}{d\lambda} = \left[ \gamma_u (\eta_u^+ - \eta_u^-) - c \right] - \frac{D N^2}{2(k\lambda - D)^2}. \tag{160}$$

At $\lambda = 1$,

$$\left. \frac{d\Pi^*}{d\lambda} \right|_{\lambda=1} = \left[ \gamma_u (\eta_u^+ - \eta_u^-) - c \right] - \frac{D N^2}{2(k - D)^2}. \tag{161}$$

By condition equation 61,

$$c > \gamma_u (\eta_u^+ - \eta_u^-) - \frac{D N^2}{2(k - D)^2}. \tag{162}$$

Therefore,

$$\left. \frac{d\Pi^*}{d\lambda} \right|_{\lambda=1} < 0. \tag{163}$$

Since the derivative is continuous in a neighborhood of $\lambda = 1$, there exists some $\bar{\lambda} \in (0, 1)$, sufficiently close to 1, such that

$$\left. \frac{d\Pi^*}{d\lambda} \right|_{\lambda=\bar{\lambda}} < 0. \tag{164}$$

Therefore, $\Pi^*(\lambda)$ is not monotonically increasing on $(0, 1)$. This proves Proposition 2.

## C  Sensitivity of Numerical Conclusions

This appendix reports a compact sensitivity check for the numerical conclusions in Section 5. The goal is to separate analytical implications from calibration-dependent patterns. The screening boundary is analytical: normal separation is feasible when $\phi_H(1 - \eta_H) < \phi_L(1 - \eta_L)$, while reverse screening is feasible when $\phi_L(1 - \eta_L) < \phi_H(1 - \eta_H)$. This comparison depends on the type-specific sanction-exposure terms and is independent of the participation cost $c$, verification congestion parameter $k$, and enforcement cost parameter $\alpha$. By contrast, the closed-form optimal policies and the non-monotonicity result depend on the quadratic verification and enforcement cost specification.

The sensitivity analysis varies $\eta_H$, $\eta_L$, $\phi_H$, $\phi_L$, $c$, $k$, and $\alpha$ around the displayed calibration. For each parameter vector, the code computes the feasible screening regime, the relaxed screening policy, the pooling benchmark, no participation, and the globally selected policy over a grid of high-type population shares $\lambda$. The exercise is not intended as calibration to deployed systems; it is a robustness check within the stylized model.

Table 4 summarizes the grid. The helpfulness values are centered around the micro-calibrated estimates reported in Section 5. The value parameters are fixed at $\gamma_v = 3.4$ and $\gamma_u(\eta_u^+ - \eta_u^-) = 0.4$. For each parameter vector, $\lambda$ is evaluated on an evenly spaced grid from 0.05 to 0.95.

Table 4: Parameter grid for the sensitivity analysis.

| Parameter | Grid values |
|---|---|
| $\eta_H$ | 0.296, 0.396, 0.496, 0.596 |
| $\eta_L$ | 0.114, 0.194, 0.274 |
| $\phi_H$ | 0.10, 0.30, 0.70, 1.20 |
| $\phi_L$ | 0.20, 0.80, 1.50 |
| $c$ | 0.15, 0.35, 0.55 |
| $k$ | 2.0, 5.5, 8.0 |
| $\alpha$ | 0.5, 0.8, 1.5 |
| $\lambda$ | 91 points from 0.05 to 0.95 |

Table 5: Sensitivity summary across the parameter grid.

| Outcome | Count | Share |
|---|---|---|
| Total parameter vectors | 3888 | 1.000 |
| Normal-separation feasible | 2862 | 0.736 |
| Reverse-screening feasible | 1026 | 0.264 |
| Boundary cases | 0 | 0.000 |
| Prop. 2 sufficient condition holds | 819 | 0.211 |
| Conditional screening value non-monotone in $\lambda$ | 2102 | 0.541 |
| Globally selected value non-monotone in $\lambda$ | 1306 | 0.336 |
| Pooling selected for some $\lambda$ | 2320 | 0.597 |
| No participation selected for some $\lambda$ | 786 | 0.202 |
| Normal screening selected for some $\lambda$ | 2280 | 0.586 |
| Reverse screening selected for some $\lambda$ | 176 | 0.045 |
| Positive-screening policy selected for some $\lambda$ | 2096 | 0.539 |
| Relaxed-boundary screening selected for some $\lambda$ | 360 | 0.093 |

Table 5 reports the main results. The screening boundary behaves as predicted by Theorem 4.1: feasible screening is classified by the analytical comparison between $\phi_H(1-\eta_H)$ and $\phi_L(1-\eta_L)$. The non-monotonicity pattern is more conditional. It appears in a substantial subset of parameter vectors, but it is not universal and can disappear once pooling and no participation are included in the global policy comparison.

The sensitivity check supports three conclusions. First, feasible screening is governed by the exposure comparison $\phi_t(1-\eta_t)$, rather than by the cost parameters $c$, $k$, and $\alpha$. Second, non-monotonicity is more common in the conditional screening value than in the globally selected value: 54.1 percent of parameter vectors exhibit a non-monotone conditional screening value, while 33.6 percent exhibit a non-monotone globally selected value. Third, pooling and no participation are economically relevant alternatives, appearing for some $\lambda$ in 59.7 percent and 20.2 percent of parameter vectors, respectively. These results reinforce the interpretation of the inverted-U path as a conditional mechanism generated by the quadratic-cost environment, not as a universal prediction.

Figure 4 visualizes the same summary. It highlights the distinction between conditional non-monotonicity in the screening value and non-monotonicity after global policy selection.

Figure 5 reports the share of globally non-monotone paths across helpfulness values. The pattern is not tied to a single exact micro-calibration, although it remains parameter-dependent.

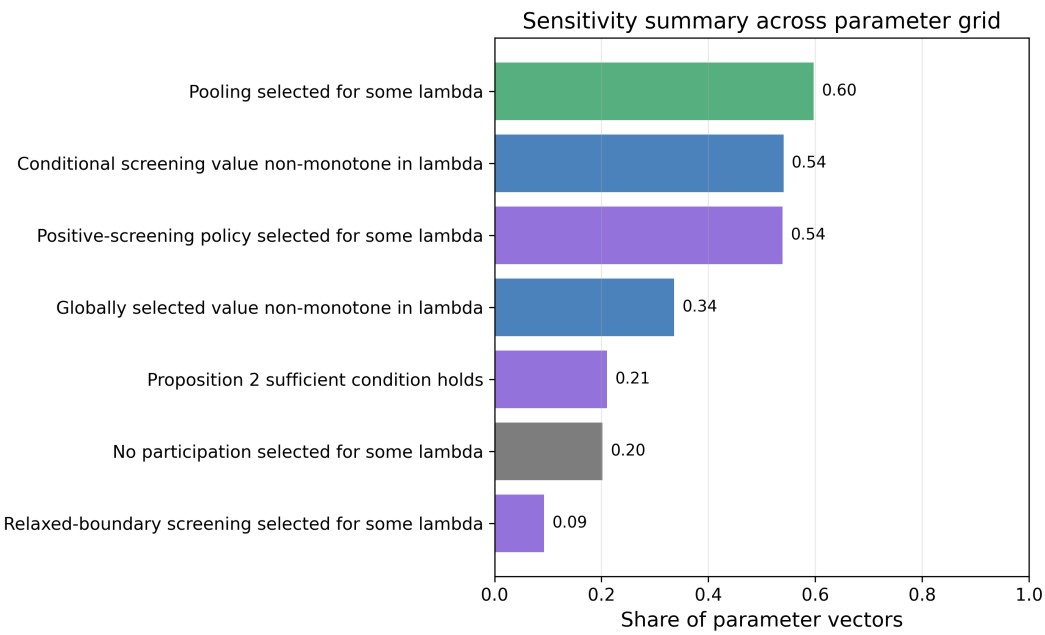

Figure 4: Sensitivity summary across the parameter grid. The inverted-U pattern appears in a substantial subset of cases but is not universal once pooling and no participation are included in the global policy comparison.

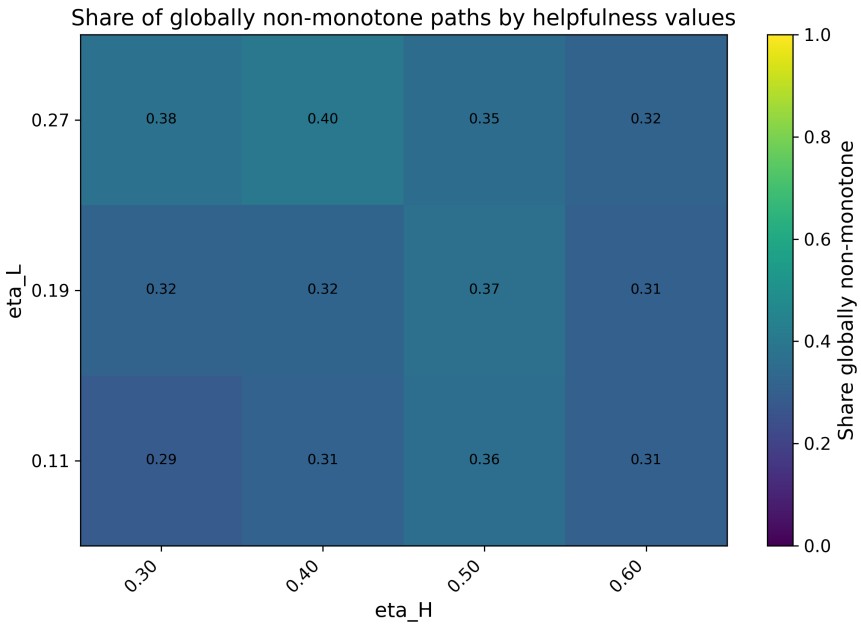

Figure 5: Share of globally non-monotone selected-value paths across helpfulness values. The figure varies $\eta_H$ and $\eta_L$ around the micro-calibrated estimates.

## D   Simulation Reproducibility Details

This appendix reports the implementation details for the illustrative simulations in Section 5. The exercise is intended to visualize the theoretical mechanisms rather than to empirically validate deployed LLM feedback systems. The accompanying Python script uses a central configuration panel, a fixed random seed, and exports all figures and CSV files used in the paper.

### D.1 Micro-Simulation Protocol

The micro-simulation uses a smoothed bigram environment to estimate type-specific helpfulness probabilities. Table 6 reports the main implementation choices. When available, the simulation uses the Brown Corpus (Kučera & Francis, 1967); otherwise, it falls back to a synthetic template corpus generated by the script. In the Brown Corpus case, sentences are shuffled using the fixed random seed before the train-validation split.

Table 6: Micro-simulation protocol.

| Item | Value |
| --- | --- |
| Random seed | 42 |
| Primary corpus | Brown Corpus |
| Fallback corpus | Synthetic template corpus |
| Training sentences | 4000 |
| Validation sentences | 1000 |
| Vocabulary size | 8000 |
| Minimum token frequency | 2 |
| Bigram smoothing | 0.01 |
| Number of degraded prompts | 100 |
| Candidate-prompt threshold | Maximum next-token probability above 0.03 |
| Special tokens excluded from degradation | [START], [END], [UNKNOWN] |
| Feedback update weight | 0.20 |
| Monte Carlo iterations | 10000 |
| High-type toy correctness probability | 0.88 |
| Low-type toy correctness probability | 0.38 |
| Helpfulness criterion | Positive validation log-likelihood improvement |

A prompt is a previous token, and a feedback item is a proposed next token. To create opportunities for improvement, the simulation first trains a smoothed bigram model and then degrades a fixed set of prompts by reducing the probability of the most likely continuation and increasing the probability of a less likely continuation. Each Monte Carlo iteration samples one degraded prompt and draws one high-type and one low-type feedback item for that same prompt. The high-type and low-type correctness probabilities, 0.88 and 0.38, are toy-generator parameters; the economic model uses the estimated helpfulness probabilities $\widehat{\eta}_H$ and $\widehat{\eta}_L$.

### D.2 Update and Validation Rule

For a prompt token $x$ and submitted feedback token $y$, the simulation updates only the flawed transition row for $x$. Let $f_{\text{old}}(\cdot \mid x)$ denote the flawed row and let $\omega = 0.20$ be the update weight. The updated row is obtained by adding $\omega$ to the submitted token and renormalizing:

$$\tilde{f}_{\text{new}}(z \mid x) = f_{\text{old}}(z \mid x) + \omega \mathbf{1}\{z = y\}, \tag{165}$$

$$f_{\text{new}}(z \mid x) = \frac{\tilde{f}_{\text{new}}(z \mid x)}{\sum_{z'} \tilde{f}_{\text{new}}(z' \mid x)}. \tag{166}$$

Let $C_{\mathcal{V}}(x, z)$ be the validation bigram count for transition $x \to z$, and let $|\mathcal{V}|$ be the total number of validation bigrams. The validation improvement is

$$\Delta\ell(x, y) = \frac{1}{|\mathcal{V}|} \sum_z C_{\mathcal{V}}(x, z) \left[\log f_{\text{new}}(z \mid x) - \log f_{\text{old}}(z \mid x)\right]. \tag{167}$$

Feedback is classified as helpful if $\Delta\ell(x, y) > 0$. The resulting estimates in the run used in the paper are $\widehat{\eta}_H = 0.396$ and $\widehat{\eta}_L = 0.194$.

### D.3 Macro Parameters and Sensitivity Grid

The micro-simulation estimates only $\widehat{\eta}_H$ and $\widehat{\eta}_L$. All other macro parameters are illustrative platform-level primitives. Table 7 reports the values used for the regime/global-policy map and for the policy-path figure.

Table 7: Macro parameters used in the numerical illustrations.

| Parameter | Regime/global-policy map | Policy-path figure |
|---|---|---|
| $\widehat{\eta}_H$ | 0.396 | 0.396 |
| $\widehat{\eta}_L$ | 0.194 | 0.194 |
| $\gamma_v$ | 4.0 | 3.4 |
| $\gamma_u(\eta_u^+ - \eta_u^-)$ | 0.4 | 0.4 |
| $k$ | 2.0 | 5.5 |
| $\alpha$ | 0.8 | 0.8 |
| $c$ | 0.20 | 0.35 |
| $\lambda$ | 0.5 | varies from 0.02 to 0.98 |
| $\phi_H$ | varies from 0.05 to 2.5 | 0.30 |
| $\phi_L$ | varies from 0.05 to 2.5 | 0.80 |
| Grid size | 180 by 180 | 200 points |

The representative scenario table fixes $\lambda = 0.5$. The normal-separation case uses $\phi_H = 0.30$ and $\phi_L = 0.80$. The near-boundary case sets $\phi_H = 0.70$ and computes $\phi_L$ from the screening boundary with multiplier 1.05. The reverse-screening case uses $\phi_H = 1.50$ and $\phi_L = 0.20$.

The sensitivity analysis uses the same policy formulas and is summarized in Appendix C. It varies the helpfulness, sanction-exposure, and cost parameters over the grid reported in Table 4; the raw grid is included in the supplementary material.

