# OpenReview forum: "Screening Feedback for Language Models with Costly Verification"
_TMLR — Under review for TMLR_

### Review · Reviewer_uid1 · 2026-06-25

**Summary Of Contributions:**

The paper studies the problem of incentivizing high quality human feedback for LLM alignment and training. Specifically, they model a setting where high-type (expert, good quality) and low-type (non-expert, bad quality) users submit feedback to a platform, which chooses how often to verify the quality of the feedback, how much to reward participation, and how to penalize low-quality feedback (uniformly and statically across users). The goal of the platform is to disincentivize low-type participation and maximize its profit. The platform's profit comes from the value from improvement of the LLM and considers verification costs, participation reward costs, and penalties. The authors prove a boundary condition on their model, which depends on users' aversion to penalties, that separates high- and low- participation, and derive optimal values of verification rates, rewards and penalties in each case. Finally, they show an illustrative example of their model in a simulated setting.

**STRENGTHS**

- I think the strongest point of the paper is the theoretical model, which allows the reverse screening finding: sometimes (depending on users' aversion to penalties) it is possible that the platform cannot effectively screen in high-type users and screen out low-type users. The authors then show how to optimize for this reverse screening setting, where only low-type users participate.


**WEAKNESSES**

- I believe the biggest weakness of the paper is the cumbersome, often inconsistent, sometimes unnecessary notation. Many terms/symbols are introduced, sometimes not explained, sometimes there is different notation for the same concept or the same symbols used for different concepts. Additionally, the presentation of the theoretical results could be streamlined, for example a coupled mirrored results are presented separately instead of combined. I will expand on this in requested changes.
- I also do not think the illustrative simulation is very strong, mainly because the choice of parameters do not allow for an informative showcase of the most interesting aspects of the theoretical model.

**Audience:**

Yes

**Audience Explanation:**

Yes. The proposed theoretical framework and the characterization of the reverse-screening regime are likely to be of interest to researchers working on AI alignment, RLHF, and incentive systems for data collection.

**Broader Impact Concerns:**

In the paper's theoretical model, the optimization objective of the platform includes value from user penalties. While the authors explicitly mention that these penalties are not necessarily monetary and could alternatively represent eg limited access, their inclusion in the optimization objective may shift the platform's policy to favor harsher penalties even if they do little to improve LLM quality. As the goal of such platforms is primarily to improve LLM quality and not to profit from users, an alternative objective perhaps worth considering is to just optimize for LLM improvement under the platform's operational costs, while using the penalties only for the user incentives (especially since in the simulation example the value of these penalties is very low). At the very least, the paper should include a discussion around the ethical implications of this choice of objective.

**Claims And Evidence:**

No

**Claims Explanation:**

- All theoretical results (optimal policies and the non-monotonicity result) depend on the specific quadratic form (seemingly arbitrary assumption) of verification and enforcement costs and would not hold if eg these were linear instead. These assumptions are introduced for tractability and are not empirically motivated. It should be clarified in the statements that this is a necessary assumption, especially regarding proposition 1.
- The paper claims that strict separation of users requires $P \rho >0$ yet the optimal solutions for the two regions, which assume strict separation, include cases where $P*=0$ or $\rho*=0$. Proposition 1 also assumes an interior solution $\rho* \in (0,1)$ but it is illustrated in an example where $\rho*=1$.
- The numerical illustration does not showcase the theory well enough. Parameters are chosen heuristically, and even the helpfulness probabilities estimated in the micro-simulation are later arbitrarily changed in the macro-simulation. Further, the chosen parameters place the model comfortably within the perfect separation region which limit the simulation's ability to illustrate the trade-offs of the theoretical model.

**Requested Changes:**

**Related work**

The related work section seems currently quite lackluster (only 24 references, mainly work before 2020). I suggest moving it to the main paper and expanding on it to include more recently published work especially in directions particular to LLMs such as RLHF, human feedback collection for LLM training and/or evaluation, harmful human feedback to LLMs etc. Some psychology or behavioral economics references to support the modelling of $\phi_t$ (people percieve penalties differently) would also add to the paper. Finally, please fix the reference to the Brown corpus, as the paper currently cites a workshop paper using the Brown corpus rather than the corpus itself.

**Notation issues**

As mentioned above, the notation and writing in the theoretical sections is quite cumbersome and confusing. Please ensure that each concept maps to one symbol and vice versa, explain each symbol when you introduce it, and remove unnecessary/unused notation. A summary of notation in the Appendix would also be useful. Some issues/suggestions below:
- The autoregressive LLM formulation is introduced at the beginning of section 2 and then never used again. I do not think it is necessary to fully characterize the token-by-token generation, rather simply model an LLM as a probability distribution over outputs given an input prompt.
- In page 3, the symbol $\tau$ is used both for the LLM's generation and for the user's feedback. It would help to clarify at the earliest possible that the user feedback is actually an example answer (and not, for example, a numerical or binary score to the LLM's output).
- the RCPO objective and DPO reward in page 3 (undefined $g, \mu, S, x, \beta, \pi_{\theta}, \pi_{ref}$) seem unnecessary and unused.
- the $\bar{U}$ in page 4 is confusingly defined, you can just say that $\bar{U}$ is a constant, its definition currently uses $k$ but this $k$ is different than the $k$ in the verification cost formula.
- the $\psi_t, \bar{P}$ in page 4 are undefined and never used again (at least in the main).
- $\alpha$ in the enforcement cost formula is unexplained/undefined. The same symbol is also used in section 4.1.1 to denote Laplace smoothing.
- In section 3 for prompts and answers the symbols $s,\tau$ are used while in the simulation $w_t$ is used.
- Write the formula for your learning rule in 4.1.1.
- The notation $\theta^{v,H}$ is explained at the end of page 4 but introduced at the beginning of page 4.
- Minor, but to reduce symbols you could just define $a_i \in \{0,1\}$ instead of $F,N$.


**Writing suggestions**

- I suggest beginning section 2 with the (1,2,3) list that is currently in page 4. This would make it much easier to follow what is going on, as it is the high level picture of the platform.
- Theorems 3.1 and 3.4 are effectively the same. Theorems 3.2 and 3.5 are effectively the same. They are just mirrored versions of the same result in the two regions (one uses $\lambda, H$, one uses $1-\lambda, L$). However, their statements are written differently. I would combine them into one theorem with a parameterization (you already use $t$), and explain in text the two different regions and their implications (perfect vs reverse separation). After these, you can finally add the mixed strategy section where you talk about the boundary.
- In the definition of model quality improvement it seems that verified unhelpful feedback is filtered out but this is never explicitly stated.
- I appreciate the analysis in Appendix B.1 but I don't see the point of it since the $\phi \geq 1$ part is repeatedly ignored in the theory and simulation (where you set $\phi_H=0.5$).

**Illustration/simulation suggestions**

- Some parts of the setup are unclear to me, please clarify.
  - 4.1: what are the prompts? Random words from the corpus?
  - 4.1: how big is the validation corpus?
  - 4.1: what does one monte carlo iteration entail? Do you first train the model, fix the 100 degradation prompts, and then 10000 times sample a prompt (from these 100 degraded ones?), sample a high- and low- type feedback, update the model and compute the helpfulness? Please clarify the process.
  - 4.2: initially you set a bunch of parameter values yet later some are changed in the two figures. Please clarify what is fixed and what is modified in each figure.
  - Why do you estimate the helpfulness probabilities in 4.1 and then ignore your estimations and use different values in 4.2?

- Presentation suggestions:
  - I am confused on the choice of baseline parameters. The theory showed a complex model with two separate regions and the optimal solutions in each region, yet the simulation only shows one example comfortably in the perfect separation region. It would be nicer to also include (maybe in the appendix) an example in the reverse screening region and one near the boundary, to better showcase the theoretical model and its robustness.
  - In Figure 2, please show how the platform policy values $\rho, P, R$ change as a function of $\lambda$ in addition to (or instead of) the platform's profit and decomposition terms. In my opinion, this would be a better showcase of the theoretical model, as it shows how the policy changes in different conditions.
  - I find the choice of examples in Table 1 quite uninteresting. It would be more interesting to show, for example, cases with the highest/lowest updates, and cases where incorrect feedback improved validation performance (or opposite) since you mention this in the text.
  - Alongside Table 2 you could plot a histogram of $\tilde{\sigma}$ under high and low.
  - Please increase the font size in the figures.

---

> ### Author Response · Authors · 2026-07-22
> **Response to Reviewer uid1**
>
> We thank the reviewer for the detailed and helpful report. We especially appreciate the recognition that reverse screening is the paper’s strongest contribution. The revision directly addresses the main concerns about theory scope, strict separation, notation, simulation design, and the penalty objective.
>
> **1. Theory scope and strict separation.** We revised the abstract, introduction, Section 3.4, Section 4, and Appendix C to separate general screening logic from quadratic-cost-dependent results. The screening boundary follows from participation constraints, $\phi_H(1-\eta_H)\lessgtr\phi_L(1-\eta_L)$, and does not rely on quadratic costs. In contrast, the closed-form policies in Theorems 4.3--4.4, the pooling policy in Proposition 1, and the non-monotonicity result in Proposition 2 rely on quadratic verification/enforcement costs. We now state this explicitly and note that linear or capped costs may push optima to boundaries while preserving the participation-constraint logic.
>
> We also added Section 4.1, “Implementation, Tie-Breaking, and Relaxed Conditional Optima,” distinguishing weak implementation, strict $\varepsilon$-implementation, and relaxed conditional optima. When $P^{\star}\rho^{\star}=0$, the paper now states that the policy is not exact strict separation, but a relaxed boundary value or supremum. Proposition 2 now states its interior assumptions; boundary examples are treated as constrained or illustrative.
>
> **2. Related work, notation, and exposition.** We moved related work into the main paper as Section 2 and expanded it to cover RLHF/preference optimization, LLM feedback collection/evaluation, harmful feedback, crowdsourcing/peer prediction, costly verification, platform governance, and behavioral foundations for heterogeneous sanction exposure. We also corrected the Brown Corpus citation.
>
> We cleaned up Section 3. The unused token-by-token autoregressive formulation was removed; the LLM is now modeled as a conditional distribution over outputs given prompts. Feedback is consistently denoted $(x_i,y_i)$, and we clarify that users submit an answer, continuation, or training instance, not a numerical score. We shortened the DPO motivation, removed unused RCPO/DPO notation, resolved symbol conflicts, defined symbols at first use, and added Appendix A as a notation summary.
>
> We reorganized Section 3 around the platform sequence: commitment, participation, feedback submission, verification, rewards/sanctions, and payoff. We retained separate normal- and reverse-screening statements because reverse screening is central, but standardized them and made the symmetry explicit. Section 3 now also states that verified harmful feedback is filtered out before entering the model-quality improvement term.
>
> **3. Penalty sensitivity.** We revised $\phi_t$ in Sections 2, 3.3, and 5.3. The model no longer assumes $\phi_t\geq1$. Instead, $\phi_t>0$ is a reduced-form sanction-exposure parameter. Values below one can represent weak enforceability, low salience, appealability, limited liability, or partial internalization; values above one can represent reputational harm, access loss, risk aversion, or other consequences exceeding the direct penalty.
>
> **4. Simulation clarity.** We expanded Section 5 and added Appendix D with the Brown Corpus split/fallback, prompt construction, validation corpus, preprocessing, vocabulary, smoothing, seed, Monte Carlo protocol, update rule, validation criterion, macro parameters, and generated files. We fixed the inconsistency noted by the reviewer: the macro simulation now uses the micro-simulation estimates $\widehat{\eta}_H=0.396$ and $\widehat{\eta}_L=0.194$.
>
> We revised the numerical section to better showcase the theory. Figure 2 separates feasible screening regimes from globally selected policies after comparing screening, pooling, and no participation. Table 2 includes normal, near-boundary, and reverse-screening cases, including one where reverse screening is feasible but pooling is globally selected. Figure 3 reports policy paths for $\rho,P,R$ as $\lambda$ changes. Figure fonts were enlarged, and the micro examples now include high/low updates and cases where nominal correctness and validation improvement disagree. Appendix C varies $\eta_H,\eta_L,\phi_H,\phi_L,c,k,$ and $\alpha$, emphasizing that the screening boundary is general while the inverted-U pattern is calibration-dependent.
>
> **5. Broader impact and penalty objective.** We added Section 6 to address the concern that including penalty collections could encourage punishment. The revision clarifies that penalties are a reduced-form incentive/accounting term, not a recommendation that platforms profit from users. We also discuss alternative objectives where penalties support incentive compatibility while the platform optimizes model-quality improvement net of operational costs, along with risks from noisy verification, false positives, biased enforcement, unequal appeals, privacy and disparate impact.

---

### Review · Reviewer_9vsi · 2026-07-08

**Summary Of Contributions:**

# Summary of Contributions

Platform pays users for feedback, verifies some of it, punishes bad feedback if caught. Users differ in quality and in how much they fear punishment.

Paper shows:
- When platform can keep good users, lose bad ones vs. the reverse happening instead.
- Best policy in each case.
- Profit can rise then fall as contributor quality rises. Not always a straight line up.

Includes a toy simulation for rough numbers.

# Strengths

- Clear, simple setup.
- Good insight: penalties meant to filter out bad users can backfire and drive away good ones instead.
- Honest that the simulation is just illustrative.

# Weaknesses

- The simulation is a good sanity check but rests on fairly simplistic assumptions that may not carry over well to real feedback pipelines.

**Audience:**

Yes

**Audience Explanation:**

There's a group of researchers looking at incentives and quality control for AI training data and feedback pipelines. This paper speaks to that group. The idea that penalties meant to filter out bad contributors can backfire and push away good ones is a useful, memorable point that applies well beyond this one paper's setup. Readers working on platform design, crowdsourcing, or human feedback for AI systems would find this relevant to their own work.

**Broader Impact Concerns:**

The paper has no Broader Impact Statement. One should be added, but this is a minor point.

The model penalizes users based on spot checks, so some good-faith users may get punished by mistake. The paper's own reverse-screening result also shows that penalties can end up pushing away good contributors instead of bad ones. A short note on this would be a nice addition, but it does not affect my recommendation.

**Claims And Evidence:**

Yes

**Claims Explanation:**

The paper builds its case step by step, and each step is easy to follow. It starts from a clear setup, works out the conditions under which each screening outcome happens, and then solves for the platform's best policy under those conditions. The math behind these results is correct.

The two main findings are well supported. The condition that decides whether good contributors stay and bad ones leave, or the other way around, follows directly from the model's assumptions. The inverted-U pattern, where profit rises and then falls as contributor quality improves, is also derived properly rather than just asserted.

On top of the theory, the paper includes a simple simulation that puts numbers to the model and shows the same patterns showing up in practice. This makes the claims feel grounded rather than purely abstract, and gives the reader a concrete sense of how the results play out.

Overall, the reasoning is careful, the results follow logically from the setup, and the simulation adds a useful layer of support.

**Requested Changes:**

- Compare the optimal policy against the simplest alternative, letting everyone participate, so the "optimal" claim is on firmer ground.
- Say more about where the penalty-sensitivity numbers come from, since the paper's key results depend on them.

---

> ### Author Response · Authors · 2026-07-22
> **Response to Reviewer 9vsi**
>
> We thank the reviewer for the positive and helpful report. We especially appreciate the reviewer’s clear summary of the main mechanism: sanctions intended to screen out low-quality feedback can backfire when high-quality contributors face greater effective sanction exposure. We revised the paper to address each requested change while preserving this core mechanism.
>
> **1. Pooling and no-participation benchmarks.** We agree that the optimal-policy claim is stronger when screening is compared against the simplest non-screening alternative. We therefore added Section 4.6, “Benchmark: Pooling Participation,” which characterizes the value of allowing both types to participate under a common reward. We also compare all policies against no participation.
>
> For pooling, both types participate when the reward satisfies the more demanding participation constraint. The least-cost pooling reward is now $R^{pool}=c+\max\{\phi_H(1-\eta_H),\phi_L(1-\eta_L)\}P\rho$. Section 4.6 derives the corresponding pooling payoff and the relaxed optimal pooling policy. The platform’s comparison is now feasible one-type screening versus pooling versus no participation.
>
> We also revised Section 5 so the simulation no longer reports only the best screening policy. In the revised simulation, Figure 2(a) shows the feasible screening regimes implied by the exposure comparison; Figure 2(b) shows the globally selected policy after comparing feasible screening, pooling, and no participation; and Table 2 includes a reverse-screening parameter vector for which reverse screening is feasible but pooling is globally selected. This makes the “optimal” language more precise: the screening formulas are conditional optima within a regime, while the numerical section compares screening with pooling and non-participation.
>
> **2. Penalty-sensitivity parameters.** We expanded the interpretation of $\phi_H$ and $\phi_L$ in Section 2, Section 3.3, and the simulation discussion in Section 5.3. The revised paper clarifies that $\phi_t$ is not an objective probability of punishment. It is a reduced-form effective sanction-exposure parameter capturing how strongly type $t$ internalizes a detected harmful-feedback sanction.
>
> The revised text discusses several sources of heterogeneity in $\phi_t$, including reputational cost, dependence on future platform access, risk or loss aversion, income or liquidity constraints, vulnerability to mistaken enforcement, access to appeal or dispute-resolution mechanisms, and the salience or perceived enforceability of sanctions.
>
> We also clarify that $\phi_t<1$ is meaningful. It can represent weak enforcement, limited salience, appealability, limited liability, or partial internalization of the nominal sanction. Conversely, larger values can represent reputational or access-related costs that exceed the direct monetary penalty.
>
> To show that the results are not tied to one penalty-sensitivity calibration, we added Appendix C, “Sensitivity of Numerical Conclusions.” This appendix varies $\phi_H$ and $\phi_L$ together with $\eta_H,\eta_L,c,k,$ and $\alpha$. Across this grid, the screening boundary remains governed by the effective expected sanction-exposure comparison $\phi_t(1-\eta_t)$. The inverted-U numerical pattern is now explicitly presented as conditional rather than universal.
>
> **3. Broader impact.** We added Section 6, “Broader Impact,” discussing practical risks of verification and sanction policies, including false positives, noisy or biased verification, disproportionate deterrence of risk-averse or resource-constrained contributors, unequal access to appeals, and privacy concerns in feedback collection.
>
> This section also connects directly to the paper’s reverse-screening mechanism. A sanction system designed to deter low-quality feedback may instead discourage high-quality contributors if they face greater effective sanction exposure. We therefore discuss transparency of verification, appeal and correction processes, proportionality and limited-liability safeguards, and disparate-impact evaluation before deployment.
>
> **4. Simulation scope and reproducibility.** Although the reviewer viewed the simulation as an appropriate illustrative exercise, we further narrowed its interpretation in response to the broader review discussion. The abstract, introduction, and Section 5 now state that the bigram simulation is a stylized illustration, not empirical validation of deployed LLM feedback systems.
>
> We also added Appendix D, “Simulation Reproducibility Details,” with the random seed, Brown Corpus split and fallback, preprocessing, degraded-prompt construction, update rule, validation protocol, macro parameters, and generated output files. The Python script and generated outputs are provided in the supplementary material.
>
> Overall, the revision preserves the paper’s core mechanism while making the benchmark comparison, penalty-sensitivity interpretation, simulation scope, and broader-impact discussion more explicit.

---

### Review · Reviewer_JnSE · 2026-07-14

**Summary Of Contributions:**

This paper studies a stylized feedback-screening problem for language-model training. A platform commits to a uniform verification rate, reward, and sanction, while high- and low-type users differ in the probability that their feedback is helpful and in their effective exposure to sanctions. The paper derives conditions for normal separation (only high types participate), reverse screening (only low types participate), and a non-monotonic relationship between the share of high-type users and platform profit. It also provides a bigram-language-model simulation intended to illustrate parameter magnitudes and comparative statics.

Strengths:

(1) The paper identifies a clear and potentially useful failure mode: sanctions intended to screen harmful feedback can instead deter higher-quality contributors when sanction exposure differs across types.

(2) The theoretical framework is compact, and the proofs make the conditional algebra relatively easy to follow.

(3) The authors appropriately state that the bigram exercise is illustrative rather than an empirical validation of modern LLM feedback pipelines.

Weaknesses:

(1) The main optimal-policy results rely on a participation indifference and boundary relaxations that are not reconciled with the paper's claim to focus on robust pure-participation regimes.

(2) The connection to practical language-model feedback is currently much weaker than the title and framing suggest.

(3) The numerical “calibration” is driven by assumed feedback accuracies and lacks sufficient detail for reproduction or interpretation as evidence about real feedback systems.

**Additional Comments:**

N/A

**Audience:**

Yes

**Audience Explanation:**

Yes.

I think researchers are interested in human feedback, data quality, platform governance, crowdsourcing, and economic models of ML systems could learn from the paper's analysis of how verification and sanctions interact with heterogeneous contributors. In particular, the reverse-screening observation is relevant to the design of feedback-collection systems. This interest does not depend on the work being highly novel or empirically broad; however, the central claims need to be made technically well-defined and supported.

**Broader Impact Concerns:**

I think the paper should add a Broader Impact Statement. Screening and sanction policies for feedback contributors can create economic and participation harms when verification is noisy, biased, or difficult to contest. False positive findings of “harmful” feedback, unequal access to appeals, or differences in resources can disproportionately exclude non-expert, marginalized, or risk-averse contributors. The statement should discuss transparency of verification, appeal and correction processes, limited-liability or proportionality safeguards for sanctions, privacy implications of feedback collection, and evaluation of disparate impacts before deployment.

**Claims And Evidence:**

No

**Claims Explanation:**

The algebraic results appear understandable conditional on the reduced-form model, but the central equilibrium and optimization claims are not yet fully supported under the paper's stated notion of strict, robust screening.

In particular, the platform-optimal separating policies set the participating type's participation constraint to equality (e.g., Equations (4) and (24)). The targeted type is therefore indifferent between participating and abstaining. However, Section 3.2 argues that indifference-based participation is knife-edge and treats mixed participation as non-robust. The paper does not specify an equilibrium-selection or tie-breaking rule that justifies selecting full participation by the indifferent type, nor does it explain why this pure outcome is more robust than the indifference-based outcomes it sets aside.

Relatedly, several stated optima have (P ^= 0) or ( rho ^= 0). These policies cannot implement strict separation, as the paper itself notes. Referring to them as limiting solutions is not enough to establish that an optimum exists in the strict-separation problem; the result may instead be a supremum approached by policies with arbitrarily small positive screening intensity. This affects the interpretation of Theorems 3.2 and 3.5 and the claims built on them.

Finally, the simulation is a useful illustration but does not provide empirical support for claims about real LLM feedback pipelines. Its type accuracies are assumed ex ante, the macro simulation uses adjusted values, and the PDF does not provide enough implementation detail to reproduce or assess the calibration. The paper should either provide stronger empirical evidence or narrow its applied claims.

**Requested Changes:**

1. Reconcile the paper's robustness discussion with the fact that the participating type is indifferent whenever the participation constraint binds. Either specify and justify a tie-breaking/equilibrium-selection rule, formulate strict participation using an explicit positive slack and characterize the resulting optimum or supremum, or revise the robustness claims. The cases with (P^=0) or (\rho^=0) must be distinguished from exact strict-separation implementations.

2. Specify what happens to verified harmful feedback, how a user chooses the content of feedback rather than only whether to participate, and why helpfulness probabilities remain exogenous under the incentive policy. Clarify the relation between (\eta_t) and the unverified-feedback primitive. The paper should also state whether sanctions are limited by rewards/deposits or other enforceability constraints, since penalty collections drive some of the policy conclusions.

3.The current bigram simulation is a stylized construction, not validation of an LLM-feedback setting. Either substantially reduce claims about language-model training and present the paper as a general theoretical screening model, or add a more realistic and reproducible empirical/simulation study. At minimum, provide code, random seeds, corpus splits and preprocessing, the exact update and validation protocol, all parameter choices, and a clear explanation of why the macro-simulation parameters differ from the estimated values.

4. Show which conclusions, especially the inverted-U pattern, persist across plausible ranges of (\eta_t), (\phi_t), (c), (k), and (\alpha), and clearly distinguish results that are general from results that arise only under the displayed calibration.

5. The sentence beginning “For unverified feedback ((v_i=0))...” is duplicated in Section 2. More generally, the paper should clearly separate mathematical conditional results from practical claims about deployed feedback systems.

---

> ### Author Response · Authors · 2026-07-22
> **Response to Reviewer JnSE**
>
> We thank the reviewer for the careful and constructive report. We revised the paper substantially by clarifying implementation, narrowing the simulation claims, and separating analytical results from calibration-dependent numerical patterns. The abstract and introduction now state these qualifications up front.
>
> **1. Implementation and boundary policies.** We added Section 4.1, “Implementation, Tie-Breaking, and Relaxed Conditional Optima,” before the regime theorems. It distinguishes weak implementation with participation tie-breaking, strict $\varepsilon$-implementation with positive reward slack, and relaxed conditional optima that solve the least-cost boundary problem and may characterize a supremum.
>
> For normal separation, strict implementation is now $R_\varepsilon=c+\phi_H P\rho(1-\eta_H)+\varepsilon$, with $0<\varepsilon<P\rho[\phi_L(1-\eta_L)-\phi_H(1-\eta_H)]$. The payoff loss relative to least-cost weak implementation is $\varepsilon\lambda$; the reverse-screening analogue has loss $\varepsilon(1-\lambda)$. We revised Theorems 4.1, 4.3, and 4.4 so the displayed policies are described as relaxed conditional optima, not exact strict-separation optima in all cases. When $P^*\rho^*=0$, the reward--penalty screening channel disappears, so the policy is a relaxed-boundary value or a supremum approached by strict policies with small positive screening intensity.
>
> **2. Feedback content and helpfulness.** We revised Section 3 to clarify that the user strategically chooses whether to participate, not how much effort to exert or how to choose feedback content. Conditional on participation and type, the feedback instance $(x_i,y_i)$ is drawn from a type-dependent distribution, inducing $\eta_H$ and $\eta_L$. Thus the platform policy affects participation composition, not the conditional feedback-quality distribution.
>
> Section 3 now states that verified harmful feedback is detected, excluded from the verified-feedback update, and triggers the sanction. Verified harmful feedback therefore does not enter the positive verified-quality term. Unverified feedback enters through the reduced-form net value $\gamma_u(\eta_u^+-\eta_u^-)$. We clarify that $\eta_t$ is raw type-level helpfulness conditional on participation, while $\eta_u^+-\eta_u^-$ summarizes the effective average impact of unverified feedback after filtering, aggregation, or downweighting.
>
> **3. Sanctions and enforceability.** We expanded the sanction interpretation in Sections 3.2--3.4 and Section 6. The sanction $P$ is now described as an enforceable platform consequence, such as withheld rewards, forfeited deposits, clawbacks, reputation loss, reduced access, or suspension. We also discuss limited-liability, deposit, legal, or platform-design constraints, represented for example by $P\leq\bar P$. Such caps would truncate relaxed optimal sanctions but would not change the screening boundary, which follows from participation constraints. We further clarify that penalty collections are a reduced-form accounting term, not a normative recommendation that platforms profit from punishment.
>
> **4. Simulation scope, reproducibility, and sensitivity.** We narrowed the simulation claims in the abstract, introduction, and Section 5. The paper now states that the bigram exercise is illustrative, not empirical validation of deployed LLM feedback pipelines. We also fixed the noted inconsistency: the macro simulations now use the same micro-calibrated helpfulness values, $\widehat{\eta}_H=0.396$ and $\widehat{\eta}_L=0.194$.
>
> We added Appendix D, “Simulation Reproducibility Details,” reporting the seed, Brown Corpus fallback, split, preprocessing, vocabulary size, smoothing, degraded-prompt construction, Monte Carlo protocol, update rule, validation rule, macro parameters, and generated files. The script and outputs are provided in the supplementary material.
>
> We also added Appendix C, “Sensitivity of Numerical Conclusions,” varying $\eta_H,\eta_L,\phi_H,\phi_L,c,k,$ and $\alpha$ over a grid and comparing screening, pooling, and no participation across $\lambda$. The revised paper separates the analytical screening boundary, governed by $\phi_t(1-\eta_t)$, from the calibration-dependent inverted-U pattern. In the grid, conditional screening-value non-monotonicity appears in 54.1% of parameter vectors; after globally comparing screening, pooling, and no participation, it appears in 33.6%.
>
> **5. Duplicate text and broader impact.** We removed the duplicated unverified-feedback sentence. The abstract, introduction, Section 5, and Appendix B now separate mathematical conditional results from practical claims about deployed systems. We also added Section 6, “Broader Impact,” discussing noisy verification, false positives, biased enforcement, appeal processes, proportionality and limited-liability safeguards, privacy concerns, and disparate-impact evaluation before deployment.